# Recent Advances in Photodetectors Based on Two-Dimensional Material/Si Heterojunctions

Yiyang Wei, Changyong Lan * , Shuren Zhou and Chun Li

State Key Laboratory of Electronic Thin Films and Integrated Devices, and School of Optoelectronic Science and Engineering, University of Electronic Science and Technology of China, Chengdu 611731, China
* Correspondence: cylan@uestc.edu.cn

**Abstract:** Two-dimensional (2D) materials have gained significant attention owing to their exceptional electronic and optoelectronic properties, including high carrier mobility, strong light–matter interaction, layer-dependent band structure and band gap. The passivated surface of 2D materials enables the fabrication of van der Waals (vdW) heterojunctions by integrating them with various other materials, such as nanowires, nanosheets and bulk materials. Heterojunction photodetectors, specifically those composed of 2D materials and silicon (Si), have attracted considerable interest due to the well-established processing techniques associated with Si and the excellent performance of the related devices. The hybrid dimension vdW heterojunction composed of 2D materials and Si has the advantages of excellent performance, low fabrication cost, and easy integration with silicon-based devices. It has unique advantages in the field of heterojunction photodetectors. This review provides an overview of the recent advancements in photodetectors based on 2D material/Si heterojunctions. First, we present the background and motivation of the review. Next, we discuss the key performance metrics for evaluating photodetector performance. Then, we review the recent progress made in the field of 2D material/Si heterojunction photodetectors. Finally, we summarize the findings and offer future prospects.

**Keywords:** two-dimensional material; heterojunctions; photodetectors

## 1. Introduction

Since the discovery of graphene in 2004 by Geim and Novoselove [1], graphene and other layered 2D materials have received considerable attention in recent decades [2–9]. These materials possess unique properties that differ from their bulk counterparts. Notable characteristics include layer-dependent band structures [10,11], strong light–matter interactions [12,13], spin–valley properties [14–16], abnormal topological structures and high-temperature ballistic transport [17,18]. To date, researchers have identified hundreds of 2D materials with diverse physical properties [19], ranging from superconductors and semimetals to semiconductors, insulators and topological insulators. As a result, 2D materials hold great promise for applications in electronics and optoelectronics.

Furthermore, layered 2D materials have a passivated surface without any dangling bonds. This feature enables the straightforward fabrication of heterojunctions by stacking an additional material on top of the 2D material without the need for lattice mismatch considerations [20]. Consequently, extensive efforts have been dedicated to the exploration of van der Waals (vdW) heterojunctions, including zero-dimensional/2D, one-dimensional/2D, 2D/2D and 2D/3D configurations [21–29]. These vdW heterojunctions exhibit intriguing electrical and optoelectrical properties and show promising potential for the next generation electronics and optoelectronics [30–41].

Photodetector are devices used to convert light signals into electrical signals and have a wide range of applications in optical communication, imaging and industrial security. Over time, various types of photodetectors have been developed. In particular,

Si-based photodetectors, such as Si p-n photodiodes, Si p-i-n photodiodes and Si avalanche photodiodes, are of great importance. The band gap of Si is about 1.1 eV, which sets the long wavelength limit (around 1100 nm) for Si-based photodetectors, thus restricting their ability to detect infrared (IR) light [42]. Moreover, Si-based photodetectors exhibit poor performance in the ultraviolet (UV) wavelength range due to strong surface recombination [43]. In addition, the fabrication process for Si p-n and p-i-n photodiodes is complex and costly [44]. While other types of photodetectors, such as InGaAs, HgCdTe and quantum well-based photodetectors, are available, their fabrication methods are also intricate and expensive. Therefore, there is a need to explore novel photodetector designs that offer high responsivity, fast response speed, broad spectral response and cost-effectiveness. Low-dimensional materials have important applications in photodetectors due to their excellent photoelectric properties, such as III–V nanowire-based photodetectors with good detection performance [45]. However, one-dimensional materials are difficult to grow on a large scale, and it is difficult to achieve the large-scale integration of devices. Quantum dot photodetectors often need to be fabricated based on III–V multilayer film technology, which is expensive and difficult to integrate with Si-based circuits [46]. Therefore, compared to bulk materials and other low-dimensional materials, 2D materials stand out in photodetectors because of their unique physical and optical properties.

Considering the facile integration of 2D materials with 3D materials through vdW forces and the well-established Si processing techniques, 2D material/Si heterojunction-based photodetectors have received considerable attention in recent years. An example of this is the fabrication of graphene/Si Schottky photodiodes, where graphene was transferred onto the surface of Si [47]. The photodiodes have exhibited a photodetection performance comparable to commercial Si photodiodes. Exploring the broadband light absorption of thick graphene films, the thick graphene/Si photodiode was able to detect incident light with a wavelength of 4 μm [48]. By replacing the 2D material with a Weyl semimetal, the detection range of the 2D material/Si photodiode was extended to an ultrabroad band range of up to 10.6 μm [49]. Two-dimensional material/Si heterojunction photodetectors offer several advantages over single-component photodetectors. First, the light absorption of 2D materials is low, although they have a large absorption coefficient due to the ultrathin thickness. The bulk Si significantly enhances the light absorption in 2D material/Si heterojunctions. Second, 2D materials exhibit diverse electrical properties, enabling the fabrication of Schottky diodes, p-n diodes, n-n or p-p diodes, as well as single barrier heterojunction photodetectors. Third, the unique properties of both the 2D material and Si are manifested in the heterojunction, enabling new functionalities and a broadband spectral response. Fourth, the difference in Fermi levels between the 2D material and Si results in the formation of an internal electrical field that promotes the separation of photogenerated electron–hole pairs and the efficient transport of photocarriers, thereby increasing the response speed. Fifth, when operated at zero or reverse bias, the 2D material/Si heterojunction photodetector exhibits low dark current, leading to improved specific detectivity. These intriguing advancements in 2D material/Si heterojunction-based photodetectors offer promising applications in future optoelectronic systems.

In this review, we aim to highlight recent advancements in the study of photodetectors based on 2D material/Si heterojunctions, with a specific focus on diode-like detectors. We will begin by introducing the figures of merit of photodetectors. Following this, we will delve into the recent progress made in the field of 2D material/Si heterojunction-based photodetectors. Notably, we will discuss the extensive research conducted on graphene/Si heterojunction photodetectors. In addition, we will explore the topic of transition metal dichalcogenide (TMD)/Si heterojunction photodetectors. It is worth mentioning that noble metal dichalcogenides (NMDs) are a subset of TMDs that exhibit distinct electrical and optoelectrical properties. Therefore, we will discuss NMD/Si heterojunction photodetectors separately from TMD/Si heterojunction photodetectors. Then, we will review other types of 2D material/Si heterojunction photodetectors. Finally, we will provide an outlook on

the future prospects and challenges associated with 2D material/Si heterojunction-based photodetectors.

## 2. Figures of Merit for Photodetectors

A photodetector is a device capable of converting light signals into electrical signals. In order to assess the performance of a photodetector, several key parameters will be introduced in this section.

### 2.1. Photoresponsivity

The parameter most commonly employed to evaluate the performance of photodetectors is photoresponsivity. Photoresponsivity quantifies the ability of a photodetector to convert light signals into electrical signals and is defined as [50]:

$$R_I = \frac{I_{ph}}{P} \text{ or } R_V = \frac{V_{ph}}{P} \tag{1}$$

Here, $I_{ph}$ ($V_{ph}$) represents the photocurrent (photovoltage), which is the difference between the currents (voltages) with and without light illumination. $P$ denotes the light power incident on the device. $R_I$ is referred to as the current photoresponsivity, while $R_V$ is known as the voltage photoresponsivity. Typically, light intensity or illumination intensity, denoted as $F$ with a unit of mW/cm$^2$, is utilized in these measurements. Assuming that the photosensitive area of the photodetector is $A$, the light power incident on the device can be expressed as $P = FA$. The photoresponsivity serves as an indicator of a photodetector's sensitivity to light illumination. It is important to note that photoresponsivity depends on both the wavelength and intensity of the incident light. In practical applications, a photoresponsivity that remains unaffected by light intensity is preferable.

### 2.2. Quantum Efficiency and Gain

Quantum efficiency is typically divided into two categories, namely, external quantum efficiency (EQE) and internal quantum efficiency (IQE), with EQE being the more commonly used parameter. The quantum efficiency serves as an indicator of a photodetector's sensitivity. The EQE ($\eta_{ext}$) is the ratio of the number of charge carriers generated in the circuit to the number of photons incident on the device [51,52]:

$$\eta_{ext} = \frac{I_{ph}/e}{P/h\upsilon} = \frac{R_I hc}{e\lambda} \tag{2}$$

where $e$ is the elementary charge value; $h$ is the Plank constant; $\upsilon$ and $\lambda$ are the frequency and wavelength of the incident light, respectively; and $c$ is the speed of light in a vacuum. It is important to note that a photodetector cannot fully absorb incident light due to the reflection and transmittance. The ratio of the number of photons absorbed by the photodetector to the number of incident photons is denoted as $\eta_a$. Additionally, a portion of the photogenerated electron–hole pairs recombine, which cannot be collected as electric current in the circuit. The collection efficiency $\eta_c$ is used to account for the collected carriers. The IQE ($\eta_i$) is the ratio of the number of electron–hole pairs generated to the number of photons absorbed by the device. If there is no gain in the device, the relationship between EQE and IQE can be expressed as follows:

$$\eta_{ext} = \eta_i \eta_a \eta_c \tag{3}$$

According to the definition, $\eta_i$, $\eta_a$ and $\eta_c$ are smaller than 1, and $\eta_{ext}$ is smaller than $\eta_i$, as stated in Equation (3). For high-quality photodiodes, IQE can approach 1.

Gain ($G$) is the number of charge carriers collected in the circuit, due to light illumination, divided by the number of photons absorbed by the device. If there is no recombination

loss of photogenerated electron–hole pairs ($\eta_c = 1$), the gain is equivalent to the IQE. The gain can be expressed as follows [52,53]:

$$G = \frac{\tau_e}{t_{tre}} + \frac{\tau_p}{t_{trp}} \tag{4}$$

where $\tau_e$ and $\tau_p$ are the lifetimes of photogenerated electrons and holes, respectively, while $t_{tre}$ and $t_{trp}$ are the transit times between electrodes for electrons and holes, respectively. In the case of a photodiode, the lifetime of the photogenerated carriers is equal to their transit time, and the gain is 1 (note that it is not 2 [51]). The relation between gain and EQE is given by

$$\eta_{ext} = \eta_a G \tag{5}$$

The EQE for diode-like photodetectors is typically less than 100% when no carrier multiplication is occurring [54]. However, for photoconductive detectors, the EQE can exceed 100%. In the case of photodetectors with an EQE of 100% (i.e., an ideal photodiode), the photocurrent responsivity is expressed as follows [52]:

$$R_I = \frac{\lambda}{1.24} \tag{6}$$

where $\lambda$ is the wavelength of the incident light with a unit of μm. Here, the unit of $R_I$ is A/W.

A simple model for the gain in a photoconductor with a perfect semiconductor can be described as follows: The mobility of electrons is higher than that of holes, so electrons move faster than holes under an external field. When an electron–hole pair is generated at the center of the device, the photogenerated electron is collected by the drain electrode before the hole reaches the source electrode. To maintain charge balance in the device, an electron is injected from the source electrode. This process continues until the hole reaches the source electrode or recombines with an electron, causing multiple electrons to be collected in the circuit and resulting in gain. Additionally, gain can also occur due to charge traps in an imperfect semiconductor. Figure 1a illustrates that a portion of the photogenerated holes are trapped in defects, requiring electrons to be injected from the source electrode to maintain charge balance when photogenerated electrons are collected at the drain electrode. Therefore, more than one electron is collected for each photon, resulting in a gain. However, these gain mechanisms do not occur in a photodiode detector. Figure 1b shows that the photogenerated carriers can be separated and collected by electrodes due to the built-in electrical field. Although electrons have higher mobility than holes, electrons cannot be injected from a p-type semiconductor. This is because electrons are minority carriers in a p-type semiconductor, and the contact between the p-type semiconductor and the electrode allows for Ohmic contact for holes but blocking contact for electrons. Therefore, no gain is possible, even under reverse bias voltage, as shown in Figure 1c. The reverse bias voltage only enlarges the depletion region and the electrical field within it, increasing the light absorption volume and the carrier drift velocity without carrier multiplication. The traps in the depletion region cannot induce gain due to the blocking contact. Additionally, the traps can lead to the recombination of photogenerated electrons and holes, as shown in Figure 1d, resulting in a reduced photoresponse. The presence of charge traps at the interface is responsible for the observation of a link between zero and open-circuit voltage in the current-voltage curve under light illumination, and reduced short-circuit current and reduced open-circuit voltage [55]. In summary, the built-in electrical field, the enhanced built-in electrical field under reverse bias and the presence of charge traps in the depletion region of photodiodes cannot lead to carrier multiplication. Consequently, the responsivity of photodiodes should be less than or equal to the value calculated by Equation (6).

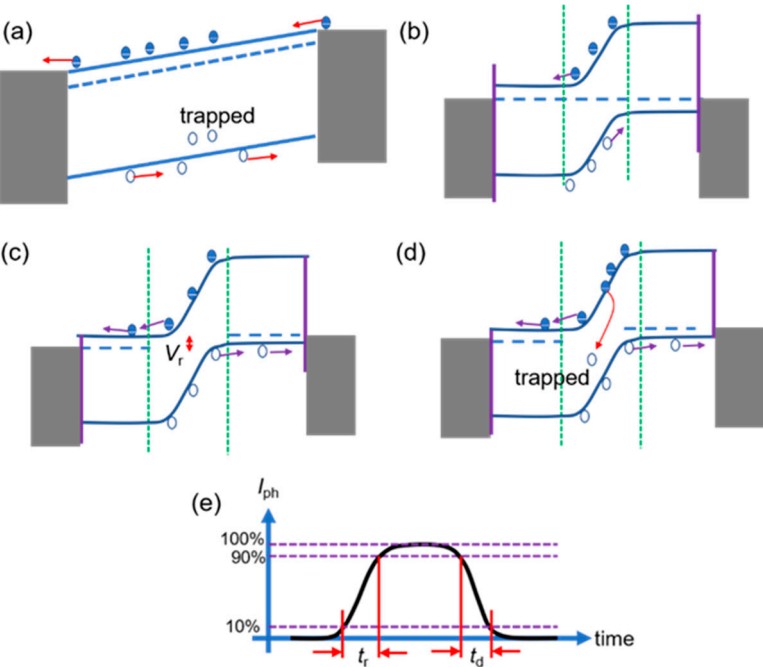

**Figure 1.** Schematic for gain and response time. (**a**) Photogenerated carrier transport in a photo-conductor. (**b**–**d**) Photogenerated carrier transport in a photodiode. (**b**) Zero bias. (**c**) Reverse bias. (**d**) Reverse bias with hole traps in the depletion region. (**e**) Response time diagram. Rise time: $t_r$. Decay time: $t_d$.

　　To induce gain in photodiodes, alternative carrier multiplication mechanisms are required. One such mechanism is avalanche breakdown, which induces impact ionization, resulting in carrier multiplication and gain in the detector. This phenomenon is widely used in avalanche photodiodes [56]. The induction of avalanche breakdown requires a large reverse bias. Recently, it has been reported that photogenerated hot carriers with sufficient energy can also lead to carrier multiplication, resulting in an EQE greater than 100% [57]. It is important to note that the photo energy must significantly exceed the band gap energy to induce impact ionization. The emergence of 2D/Si heterojunction photodetectors brings about unique properties, including gain. The gain in graphene/Si heterojunctions is well-explained and will be discussed later. However, the gain mechanism remains unclear for semiconducting 2D material/Si heterojunctions. For instance, zero bias semiconducting 2D material/Si heterojunction photodetectors have exhibited remarkably high responsivity, far surpassing the value calculated by Equation (6) [58–61]. Additionally, reverse biased semiconducting 2D material/Si heterojunctions without avalanche breakdown have also shown gain [62–66]. Therefore, further investigation into the gain mechanism in these heterojunctions is necessary.

### 2.3. Noise Equivalent Power

　　The minimum detectable signal of a photodetector is determined by its noise level. To quantify the ability to detect weak optical signals, the noise equivalent power (NEP) is commonly used. NEP is the incident light power required to achieve a signal-to-noise ratio of one within a bandwidth of 1 Hz [52]:

$$P_{NEP} = \frac{i_n}{R_I} \tag{7}$$

where $i_n$ is the square root noise current per unit bandwidth with a unit of $A/\sqrt{Hz}$. The unit of the NEP is $W/\sqrt{Hz}$.

### 2.4. Detectivity and Specific Detectivity

A photodetector with a low NEP is preferred in order to achieve optimal performance. In the field of device characterization, it is common to use a parameter that quantifies the performance of a device, where a larger value indicates better performance. However, it is not desirable to state that a smaller parameter value corresponds to better device performance. To address this, another parameter called detectivity ($D$) is defined as the reciprocal of NEP, as shown in Equation (8):

$$D = \frac{1}{P_{NEP}} \tag{8}$$

It is important to note that the detectivity value depends on the photosensitive area of the device. To facilitate performance comparisons between photodetectors with different photosensitive areas, a parameter called specific detectivity ($D^*$) is introduced, as shown in Equation (9):

$$D^* = \frac{\sqrt{A}}{P_{NEP}} \tag{9}$$

where $A$ is the photosensitive area of the device. The unit of specific detectivity is cm·Hz$^{1/2}$/W, which is also written as Jones. For devices where shot noise dominates the noise characteristics, the $D^*$ is expressed as:

$$D^* = \frac{R_I \sqrt{A}}{\sqrt{2eI_d}} \tag{10}$$

where $I_d$ is the dark current of the device. In the case of diode-like devices operating at zero bias, $D^*$ is expressed as [67]:

$$D^* = \frac{R_I \sqrt{A}}{2\sqrt{eI_{sat}}} \tag{11}$$

where $I_{sat}$ is the absolute value of the reverse saturation current. The specific detectivity provides a normalized value that takes into account both photosensitive area and bandwidth, enabling effective performance comparisons between different devices.

### 2.5. Response Time

The response time is a critical parameter that characterizes the performance of photodetectors. Typically, this parameter is represented by the rise time and decay time of a photodetector. The rise time and decay time are defined as the time required for the photocurrent to rise from 10% to 90% of its static value and vice versa, respectively, as shown in Figure 1e. The response time of a photoconductor is influenced by the lifetime of the photogenerated carriers and their mobility. In the case of diode-like photodetectors, the response time is determined by the diffusion time of carriers out of the depletion region, the drift time within the depleted region and the capacitance of the junction [51].

### 2.6. Linear Dynamic Range

In practical applications, it is crucial for a photodetector to have a linear response when converting optical signals into electrical signals in order to avoid distortion. However, at high light power levels, the photocurrent tends to saturate. To evaluate the linear response capability, the concept of linear dynamic range (LDR) is introduced. LDR represents the power range within which a photodetector maintains a linear response to incident light. It is defined as follows [68]:

$$r_{LDP} = 10 \lg \frac{P_{max}}{P_{min}} \tag{12}$$

Here, $P_{min}$ and $P_{max}$ correspond to the minimum and maximum light power levels, respectively, at which the photodetector exhibits linear response. The unit of LDR is decibel (dB). It is worth noting that most commercial photodiodes demonstrate linear response until they reach saturation. In these cases, $P_{min}$ represents the NEP without normalization to bandwidth. However, it is important to note that many photoconductive detectors reported

in the literature exhibit sublinear behavior, making LDR inappropriate for characterizing them.

It is important to emphasize that the commonly misused definition of $20 \lg(P_{max}/P_{min})$ in the literature is incorrect. In engineering, the use of 10lg is appropriate for the power range with dB units, while 20 lg is used for the amplitude range with dB units. Moreover, it is not appropriate to use photocurrent to define LDR as $20 \lg(I_{pmax}/I_{pmin})$, where $I_{pmax}$ and $I_{pmin}$ are the maximum and minimum photocurrents that the device responds linearly to incident light. This is because the focus should be on the linear response to incident light power and not on the photocurrent itself, making such a definition meaningless.

## 3. Photodetectors Based on 2D Material-Si Heterojunctions

A variety of 2D material/Si heterojunction photodetectors have been extensively studied to date. In this session, we will present the recent advancements in 2D material/Si heterojunction photodetectors. Since graphene has been the subject of intense research, we will first discuss graphene/Si heterojunction photodetectors. Subsequently, we will examine TMD/Si heterojunction photodetectors. Notably, although NMD falls under the category of TMD, we will devote a separate section to discussing NMD/Si heterojunction photodetectors due to their unique properties. Following this, we will discuss other types of 2D material/Si heterojunction photodetectors.

### 3.1. Graphene/Si Heterojunctions

Graphene, renowned for its remarkable electrical and optical properties, has emerged as the most extensively studied 2D material. In particular, graphene exhibits ultra-high carrier mobility, reaching $2 \times 10^5$ cm$^2$/Vs at low temperature [69], which greatly facilitates carrier transport. Moreover, monolayer graphene absorbs only 2.3% of the incident light within the 400–800 nm wavelength range [70]. In addition, graphene functions as a semimetal with high conductivity [71,72]. This unique combination of low light absorption and high conductivity positions graphene as a potentially transparent electrode [73,74]. Consequently, graphene can be used to fabricate Schottky diodes, which can serve as solar cells and photodetectors. In this review, we will focus only on the advancements in graphene/Si heterojunction photodetectors.

Typically, graphene on substrates exhibits unintentional p-doping [1,75,76]. Consequently, n-type Si is usually used to form a heterojunction with graphene. Given the semimetal nature of graphene, the graphene/Si heterojunction functions as a Schottky diode. An et al. performed a transfer of chemical vapor deposition (CVD)-synthesized graphene onto lightly n-doped Si (ρ = 1–10 Ω·cm) to create a Schottky photodiode, as shown in Figure 2a [47]. Their findings revealed a substantial photovoltage responsivity exceeding $10^7$ V/W and a low NEP of approximately 1 pW/Hz$^{1/2}$ when operating in photovoltaic mode. The photodiode demonstrated a current responsivity of 435 mA/W and an expansive LDR of 60 dB, as shown in Figure 2b. Furthermore, the device exhibited fast response times in the millisecond range, and the photo-to-dark current ON/OFF ratios exceeded $10^4$. However, the photodiodes displayed a small short-circuit current under light illumination, which was attributed to the limited density of states of graphene near the Dirac point. The small short-circuit current, however, was attributed to the poor interface between graphene and Si by Song et al. [55], resulting in the fast recombination of photogenerated carriers. Nevertheless, the straightforward fabrication method of these photodiodes has sparked considerable enthusiasm for further investigation of graphene/Si photodiodes.

In general, un-passivated Si surfaces possess numerous surface states that can function as recombination centers for photogenerated carriers, leading to significant leakage current. Coating the surface with a layer of graphene alone does not effectively passivate the underlying Si due to its saturated surface. Therefore, the incorporation of a passivation layer is necessary to improve the performance of graphene/Si heterojunction-based photodiodes. Li et al. introduced a thermally grown SiO$_2$ layer between Si and graphene as an interfacial layer to passivate the dangling bonds in the Si surface [77]. The introduction

of this interfacial layer resulted in a significant reduction in the dark current, while the photoresponsivity and response speed remained unaffected, as shown in Figure 2c. As a result, the NEP, specific detectivity and the photo-to-dark current ON/OFF ratio showed substantial improvements, as can be seen in Figure 2d.

The internal built-in electrical field at the graphene/Si interface is crucial for achieving high-performance photodetectors. Previous studies on graphene/Si solar cells have demonstrated that p-doping on graphene can increase both the Schottky barrier height at the interface and enhance the conductance in graphene, resulting in improved device performance [78,79]. Yu et al. successfully employed Si quantum dots (QDs) to achieve p-doping in graphene, and the resulting photodetectors based on the doped graphene-Si heterojunctions exhibited significantly enhanced performance, as shown in Figure 2e [80]. The introduction of Si QDs on top of graphene increased the built-in potential of the graphene/Si junction. This doping strategy led to remarkable achievements, including a record-high responsivity of 0.495 A/W, as can be seen in Figure 2f, an unprecedented short response time of 25 ns and an excellent specific detectivity of $7.4 \times 10^9$ Jones.

To address the issue of high light reflection (35%) in planar Si, which hinders efficient light absorption and results in low EQE, researchers have commonly used antireflective layers and nanostructured Si in graphene/Si heterojunction-based solar cells [81–83]. However, these approaches have rarely been explored in the context of graphene/Si heterojunction photodiodes. Recently, Zhao et al. proposed a design that incorporates highly anti-reflective Si nanometer truncated cone arrays (Si NTCAs) in graphene/Si heterojunction photodetectors [84]. The Si NTCAs effectively reduced light reflection compared to planar Si due to the light-trapping effect of the nanostructures. As a result of the enhanced light absorption, the graphene/Si NTCAs heterojunction photodetectors achieved an impressive EQE of 97% at 790 nm, along with a short rise/fall time of 60/105 μs.

The detection spectral range of the graphene/Si heterojunction photodetectors is primarily determined by the Si component, as graphene typically serves as the transparent electrode. However, graphene can also act as a photosensitive material. When photoexcited carriers are generated in graphene, they can be emitted to Si across the Schottky barrier, a process known as internal photoemission, enabling the detection of light with photon energy higher than the Schottky barrier [85]. Despite graphene's broadband spectral absorption capabilities, its absorption of incident light is extremely low, resulting in a weak photoresponse to light beyond the absorption edge of Si. To overcome this limitation, Casalino et al. designed a resonant cavity structure to enhance the light absorption in graphene at 1550 nm, as schematically shown in Figure 3a [86]. The spectral response of the detector without the Au back mirror is shown in Figure 3b, which shows pronounced wavelength-dependent periodic photocurrent peaks that align with the resonant characteristics of a Fabry–Perot cavity. By introducing a Au back mirror, the responsivity of the photodetector was increased by a factor of 3, accompanied by a blue spectral shift of the resonance wavelengths, as shown in Figure 3c. The photoresponsivity also exhibited a dependence on the reverse bias voltage, reaching approximately 20 mA/W at a bias voltage of 10 V near 1550 nm, as shown in Figure 3d. In addition to resonant cavity structures, other forms of graphene have been used to construct broadband graphene/Si photodiodes [87]. For instance, Peng et al. fabricated a highly crystalline macroscopic assembled graphene nanofilm (nMAG) and transferred it onto the surface of Si, forming an nMAG/Si heterojunction photodetector [48]. The 45 nm thick nMAG displayed an extinction coefficient of 74–81%, corresponding to approximately 40% light absorption, as shown in Figure 3e. Furthermore, nMAG exhibited a long carrier relaxation time (~20 ps), a low work function (4.52 eV) and suppressed carrier number fluctuations, all of which are advantageous for achieving high-performance nMAG/Si heterojunction photodetectors. Under these circumstances, the device operated in the wavelength range of 1.5–4.0 μm at room temperature Figure 3f, demonstrating a fast response time, as shown in Figure 3g, and high detectivity under pulsed laser illumination. The spectral response of the photodetector extended beyond the intrinsic absorption of Si, indicating that nMAG played a significant role in the

photoresponse. The specific photoresponse mechanism depended on the photon energy range, as illustrated in Figure 3h. When the photon energy exceeded 2.1 eV, the photogenerated electrons had sufficient energy to overcome the Schottky barrier and contribute directly to the photocurrent. Conversely, for photon energies below 2.1 eV, the photogenerated electrons thermalized into a Fermi–Dirac distribution, and only those electrons with energy above the Schottky barrier height could contribute to the photocurrent.

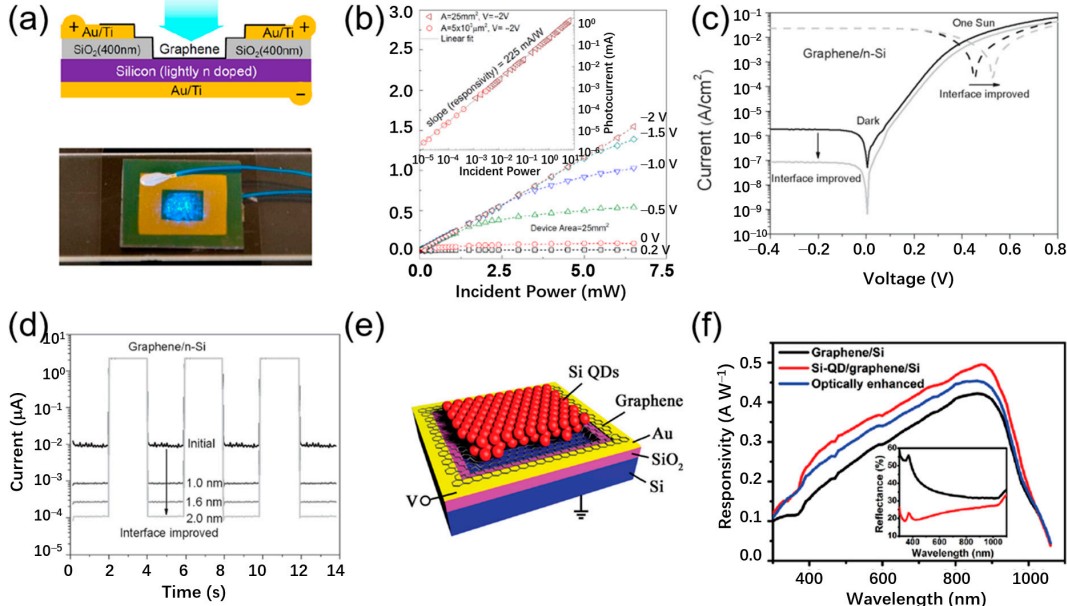

**Figure 2.** Photodetection performance of graphene/n-Si photodiodes. (**a**) Schematic of the graphene/n-Si heterojunction device and digital photograph of the device. (**b**) Photocurrent as a function of light power under different bias voltage. Inset: natural logarithm plot of the photocurrent vs. light power with bias voltage of −2 V. (**a**,**b**) Reproduced with permission from Ref. [47]. Copyright 2013, American Chemical Society. (**c**) The current-voltage curves of graphene/n-Si and the heterojunction with 2 nm-thick interfacial modification. (**d**) Oxide thickness dependent response at 890 nm. (**c**,**d**) Reproduced with permission from Ref. [77]. Copyright 2016, WILEY-VCH. (**e**) Schematic of the Si QDs doped graphene/n-Si heterojunction device. (**f**) Responsivity as a function of wavelength for devices with different structures. (**e**,**f**) Reproduced with permission from Ref. [80]. Copyright 2016, WILEY-VCH.

Unlike photoconductive detectors, which always exhibit a gain, diode-like photodetectors do not exhibit gain until ionization impacts occurs, leading to carrier multiplication. However, researchers have discovered that the EQE of the graphene/Si photodetectors consistently exceeds 100%. Initially, the photosensitive area of the graphene/Si junction was thought to be the sole contributor to the photocurrent. However, it has been proven that the area beneath the $SiO_2$ layer also contributes to the photocurrent. Riazimehr et al. employed scanning photocurrent measurements to investigate the spatial distribution of the photocurrent in the graphene/Si heterojunction [88]. They observed that the Si region away from the graphene/Si junction also contributed to the photocurrent due to the diffusion of photogenerated carriers towards the junction area. Additionally, the photocurrent generated at the graphene/$SiO_2$/Si region exhibited a strong reverse bias voltage-dependent behavior, as shown in Figure 4a. At low bias voltages, the photogenerated holes in the Si underneath the graphene/$SiO_2$ had to diffuse towards the graphene/Si junction. The presence of a large number of interfacial defects near the $SiO_2$/Si interface acted as recombination centers, resulting in a lower photocurrent. Conversely, when the bias voltage exceeded the inversion voltage of the Si underneath the $SiO_2$ layer, the recombination rates are significantly reduced due to the passivation of the interfacial states by the inversion layer, resulting in a higher photocurrent. Therefore, it is crucial to accurately estimate the

actual photosensitive area when evaluating the performance of the graphene/Si hetero-junction photodetectors. Srisonphan et al. investigated the photodetection performance of graphene/Si heterojunctions with a graphene/SiO$_2$/Si step [89]. The devices exhibited a large photoresponsivity of 1.2 A/W for p-Si and 0.45 A/W for n-Si, corresponding to EQEs of 235% and 88%, respectively. The multiplication gain was attributed to the presence of the graphene/SiO$_2$/Si step with a nanoscale gap, as illustrated in Figure 4b, which resulted in the formation of a graphene/air/Si interface acting as a nanoscale vacuum electronic structure. Impact ionization, initiated by photoinduced carrier injection into the self-induced localized electric field (reaching 10$^6$ V/cm) distributed in a 2D electron gas, as shown in Figure 4c, which was more pronounced when the graphene/Si junction area is small compared with the nanoscale gap. Additionally, p-Si exhibited a higher quantum gain than n-Si due to the higher impact ionization rate, and the gain was also dependent on the intensity of the incident light power. Bartolomeo et al. investigated photodetectors based on graphene/Si nanotip heterojunctions and achieved a high photoresponsivity of 3 A/W under white light illumination [90]. The nanotip not only enhanced light absorption, but also created a large electric field at the tip apex, which could induce impact ionization, resulting in gain. Yin et al. introduced a thin wide-bandgap AlN insulator at the graphene/Si heterojunction interface and found that the strong electrical field within the insulator caused impact ionization, which resulted in gain [91]. The Fermi level of graphene in graphene/Si Schottky diodes can be adjusted by a bias voltage due to the limited density of states of graphene near the Dirac point, resulting in a tunable Schottky potential [92]. Consequently, the barrier for electron transfer from graphene to Si can be lowered under reverse bias voltage. By exploiting this characteristic, we achieved high-performance graphene/Si heterojunction photodetectors by inserting a WS$_2$ interfacial layer between graphene and n-Si [93]. The WS$_2$ interfacial layer was directly synthesized on the Si surface through magnetron sputtering followed by sulfurization. The interfacial layer not only passivated the surface states on the Si surface but also acted as a hole-trapping layer due to its band alignment with Si. The photogenerated holes were trapped at the WS$_2$/Si interface, attracting electrons into the Si due to the reduced barrier height under reverse bias, as shown in Figure 4d, resulting in gain similar to that in a photoconductive device. In this configuration, a responsivity of 54.5 A/W was achieved at 800 nm, as shown in Figure 4e. In addition, the responsivity was found to be dependent on the size of the graphene/Si junction area, i.e., decreasing as the junction area increased, due to an increased recombination rate. Moreover, by finely tuning the interfacial layer thickness, the responsivity could be further increased to 8.96 × 10$^4$ A/W, as shown in Figure 4f [94]. Chang et al. fabricated a gate-controlled photodetector based on a graphene/Si Schottky junction [95]. The device demonstrated a substantial ON/OFF photo-switching ratio of 10$^4$, a high photoresponsivity of 70 A/W and a low dark current on the order of $\mu$A/cm$^{-2}$ across a broad wavelength range of 395 to 850 nm. As shown in Figure 4g, the device consisted of a graphene/Si heterojunction, an Al$_2$O$_3$ dielectric layer and a ZnO top gate. The authors proposed a unique gain mechanism in which the injection rate of holes is amplified by the negative gate voltage, leading to an increased photocurrent. The current observed was not a pure photocurrent but rather an amplified substrate current initiated by the photocarriers, as illustrated in Figure 4h.

For better comparison and understanding of the graphene/Si Schottky junction photodetectors, the performance parameters of some typical devices are listed in Table 1.

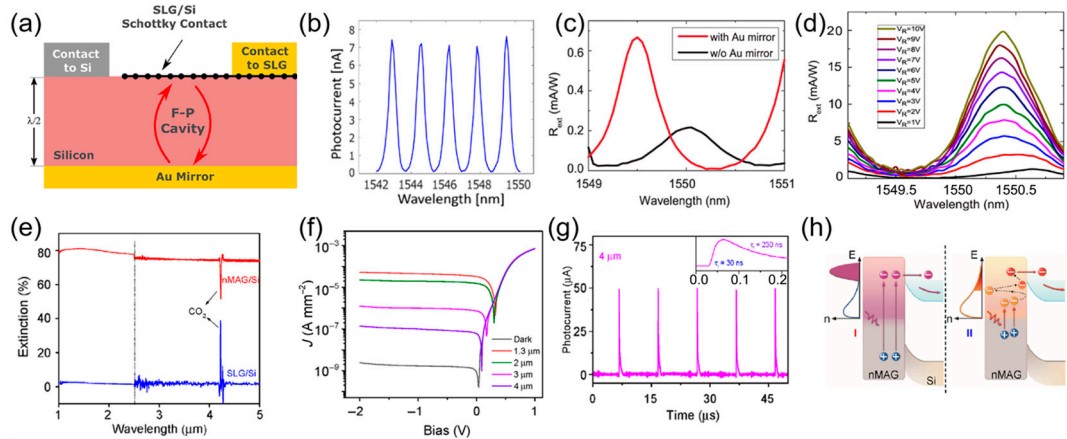

**Figure 3.** Broadband graphene-Si heterojunction photodetectors. (**a**) Schematic cross-section of the resonant cavity enhanced graphene/Si Schottky photodetector. (**b**) Spectral response of the device without backside Au mirror. (**c**) Responsivity spectra of the device with and without backside Au mirror. (**d**) Responsivity spectra under different reverse bias voltages. (**a**–**d**) Reproduced with permission from Ref. [86]. Copyright 2017, American Chemical Society. (**e**) Extinction spectra of single layer graphene/Si and nMAG/Si heterojunctions. (**f**) Current density vs. bias voltage in dark and under laser illumination at different wavelengths with light intensity of 40 mW/mm$^2$. (**g**) Photocurrent vs. time under pulsed light illumination. (**h**) Energy band diagrams for two operation mechanisms at different wavelengths. (**e**–**h**) Reproduced with permission from Ref. [48]. Copyright 2022, WILEY-VCH.

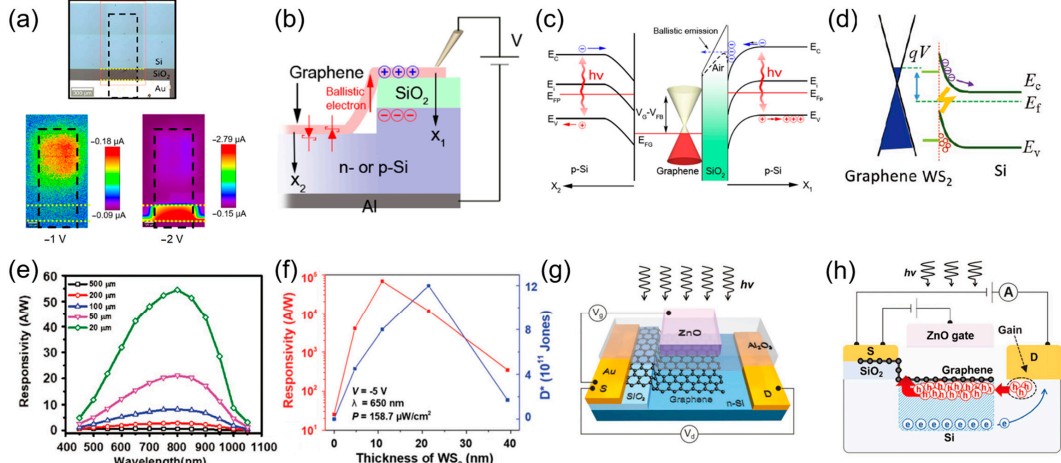

**Figure 4.** Graphene/Si heterojunction characteristics concerning gain. (**a**) Photocurrent mapping of the graphene/n-Si device under different bias voltages. Reproduced with permission from Ref. [88]. Copyright 2017, American Chemical Society. (**b**) Schematic of the cross-section of the graphene/n-Si heterojunction. (**c**) Band diagram along different directions shown in (**b**). (**b**,**c**) Reproduced with permission from Ref. [89]. Copyright 2016, American Chemical Society. (**d**) Band diagram of the graphene/WS$_2$/Si heterojunction under reverse bias and light illumination. (**e**) Responsivity as a function of wavelength for devices with different sizes. (**d**,**e**) Reproduced with permission from Ref. [93]. Copyright 2019, WILEY-VCH. (**f**) Responsivity and specific detectivity as a function of WS$_2$ film thickness. Reproduced with permission from Ref. [94]. Copyright 2021, Royal Society of Chemistry. (**g**) Schematic of the gate-tunable graphene/Si photodetector. (**h**) Schematic of the device describing photocurrent component at negative gate bias. (**g**,**h**) Reproduced with permission from Ref. [95]. Copyright 2018, WILEY-VCH.

**Table 1.** Summary of properties of graphene/silicon heterojunction devices.

| Materials | Measurement Conditions | R/A·W$^{-1}$ | D*/Jones | EQE | Time (Rise/Down) | Ref. |
|---|---|---|---|---|---|---|
| 3L Gr/Si+PCA | 885 nm/−2 V | 0.435 | | 65% | | [47] |
| Gr/n-Si | 890 nm/0 V | 0.73 | $4.08 \times 10^{13}$ (in air) $5.77 \times 10^{13}$ (in vacuum) | | 0.32 ms/0.75 ms | [77] |
| Si QDs/Gr/Si | 877 nm/−1 V | 0.495 | $7.4 \times 10^9$ | | 25 ns | [80] |
| Gr/Si NTCAs | 780 nm/0 V | 0.45 | | 97% | 60 µs/105 µs | [84] |
| Gr/Si | 1550 nm/10 V | 0.02 | | | | [86] |
| nMag/Si | 1300 nm/−1 V | | $1.6 \times 10^{11}$ | | 20 ns/200 ns | [48] |
| Gr/SiO$_2$/p-Si | 633 nm/−5 V | 1.2 | | 235% | 40 ns/100 ns | [89] |
| Gr/SiO$_2$/n-Si | 633 nm/−5 V | 0.45 | | | 40 ns/100 ns | [89] |
| Gr/Si-tip | 880 nm/−0.5 V | 3 | | 88% | | [90] |
| Gr/AlN/n-Si | 850 nm/−10 V | 3.96 | $1.13 \times 10^8$ | | | [91] |
| Gr/WS$_2$/Si | 800 nm/−0.3 V | 54.5 | $4.1 \times 10^{12}$ | | 45 µs/210 µs | [93] |
| Gr/WS$_2$/Si | 690 nm/−5 V | $8.96 \times 10^4$ | $8.86 \times 10^{11}$ | | 0.84 ms/2.1 ms | [94] |
| ZnO/Gr/Si | 850 nm/ $V_g = -15$, $V_d = 0.1$ V | 70 | $2 \times 10^{13}$ | | | [95] |

Gr: Graphene.

### 3.2. TMD/Si Heterojunctions

Except graphene, TMDs have been extensively studied as 2D materials due to their superior electrical and optoelectronics properties [96–98]. These properties include layer-dependent band gaps and band structure, large exciton binding energy, spin–valley effect, strong light–matter interaction and high carrier mobility [99–101]. When combined with Si to form van der Waals heterojunctions, the heterojunction devices can fully utilize the advantages of both materials, making them favorable for photodetection. Yim et al. reported the first few-layer MoS$_2$/Si pn photodiode, where the MoS$_2$ film was synthesized by sulfurizing a Mo film and transferred onto the surface of a p-Si substrate to form a heterojunction [102]. The photodiode showed strong photoconductivity, which could be adjusted by varying the thickness of the MoS$_2$ layer. Subsequently, significant efforts have been made to explore TMD/Si heterojunction-based photodetectors. Li et al. prepared monolayer MoS$_2$ flakes by mechanical exfoliation and transferred them onto the surface of Si substrates with different types of conductivity to form MoS$_2$/Si heterojunctions [62]. They used a conductive atomic force microscope to study the photoresponse of the heterojunction devices. Both the monolayer MoS$_2$/n-Si heterojunction and the monolayer MoS$_2$/p-Si heterojunction exhibited photodiode-like behavior, and the monolayer MoS$_2$/n-Si heterojunction showed better performance with the highest photoresponsivity of 7.2 A/W. The higher photoresponsivity of the monolayer MoS$_2$/n-Si heterojunction was attributed to the larger built-in electrical field revealed by a Kelvin force microscope. Li et al. studied the photoresponse mechanism of monolayer MoS$_2$/p-Si using a photocurrent mapping technique and found that the photosensitive region under both reverse and forward bias is the junction region [103]. They also measured the spectral response and found that it was identical to the absorption spectrum of the monolayer MoS$_2$, indicating that the photogenerated carriers were mainly present in the monolayer MoS$_2$. Their results were different to the findings reported in the literature and may be attributed to the high doping level of the Si substrates used in their study. In our study, we synthesized few-layer WS$_2$ films by sulfurizing W film and transferred them onto the surface of p-Si substrates to form

p-n photodiodes, as shown in Figure 5a [104]. We observed both Zener and avalanche breakdowns at low temperatures, while only Zener breakdown was observed at room temperature, which was conformed in few-layer $WS_2$ flake/p-Si heterojunction by another research group [105]. The device under reverse bias exhibited excellent photodetection performance, including good linearity of the photocurrent to light intensity, excellent stability, fast response and wide spectral response. The maximum responsivity was observed at 660 nm with a value of 5.7 A/W. The spectral response differed between both $WS_2$ and Si, indicating that photogenerated carriers were generated in both $WS_2$ and Si for short wavelengths and only in Si for wavelengths beyond the band edge of $WS_2$.

During the transfer of a 2D material onto the surface of a Si substrate, contamination of the 2D material/Si interface is inevitable. This contamination often leads to a degradation of the photodetector performance of the photodetector devices. Therefore, a more advantageous approach is the direct synthesis of 2D material on Si, as it can improve the performance of the junction devices. For example, Zhang et al. successfully synthesized a multilayer n-$MoS_2$ film directly on an n-Si substrate through a thermolysis method [106]. The resulting heterojunction demonstrated diode-like electrical behavior in the absence of light and exhibited excellent photodetection performance under reverse bias. At a bias voltage of $-2$ V, the responsivity reached 11.9 A/W at 650 nm. Moreover, due to the high quality of the hetero-interface, the device showed a rapid response with a rise time of 30.5 µs and a decay time of 71.6 µs. Another study by Lu et al. employed pulsed laser deposition (PLD) to deposit a $MoTe_2$ film directly on an n-Si substrate, forming an n-n junction [107]. To improve the carrier collection efficiency, a layer of graphene was transferred onto the surface of the $MoTe_2$ film. The resulting device demonstrated a clear diode-like behavior and exhibited excellent photodetection performance, including a high responsivity of 0.19 A/W and a large specific detectivity of $6.8 \times 10^{13}$ Jones. Due to the narrow bandgap of $MoTe_2$, the device was capable of detecting light with wavelengths ranging from 300 to 1800 nm, as shown in Figure 5b. Moreover, the device exhibited a fast response speed, with a 3 dB bandwidth approaching 0.12 GHz. Lei et al. fabricated a $MoTe_2$/Si heterojunction by directly growing a $MoTe_2$ film onto a Si substrate using a shadow mask-assisted CVD method [108]. The resulting device exhibited good photodetection performance, with a high photo-switching ON/OFF ratio of ~$10^4$, a zero-bias responsivity of 0.26 A/W and a high specific detectivity of $2 \times 10^{13}$ Jones at 700 nm. Wu et al. fabricated a $MoSe_2$/Si heterojunction array by directly depositing a Mo film onto a Si substrate and performing post-selenization [109]. This device demonstrated an impressive responsivity of 720.5 mA/W and a high specific detectivity of $10^{13}$ Jones at zero bias. Reactive magnetron sputtering and direct magnetron sputtering with post-annealing have also been used to directly deposit TMD films directly onto Si substrates to form heterojunctions, resulting in decent photodetection performance [65,110].

The in-plane carrier transport in TMDs is exceptional due to their large in-plane carrier mobility. However, the vdW gap between the layers can hinder carrier transport. In the case of a 2D/Si heterojunction, the carrier extraction is inefficient when the 2D plane is parallel to the junction interface. To address this issue, Wang et al. successfully achieved a vertical aligned $MoS_2$ 2D plane on a p-Si substrate using magnetron sputtering and subsequent annealing [111]. The resulting detector demonstrated sensitivity to a broad range of wavelengths from visible to near-IR light and exhibited a high detectivity of $10^{13}$ Jones. This can be attributed to the strong built-in electrical field and high-speed path for the transport of photogenerated carriers, resulting in a fast response speed with a rise time of 3 µs. Similarly, Kim et al. synthesized vertically layered $WS_2$ nanosheets on a p-Si substrate through radio frequency sputtering at 400 °C, forming a heterojunction photodetector [112]. As expected, this device demonstrated an ultrafast photoresponse with a response time of 1.1 µs. Cong et al. used the CVD method to synthesize vertically oriented few-layer $MoS_2$ nanosheets, as shown in Figure 5c, which were then transferred onto a p-Si substrate to study the lateral photovoltaic effect [113]. The device exhibited a strong lateral photovoltaic effect due to the strong light absorption and fast carrier transport

speed of the vertically aligned $MoS_2$ sheets, as well as the strong built-in electrical field at the interface. The intralayer and interlayer transport times in the vertically aligned $MoS_2$ nanosheets were determined to be 5 and 11 ns, respectively. Taking advantage of the superior properties of the vertically aligned $MoS_2$ nanosheets/Si heterojunction, Qiao et al. observed that the heterojunction device exhibited a high responsivity of 908.2 mA/W and a high specific detectivity of $1.9 \times 10^{13}$ Jones at 808 nm under a bias voltage of $-2$ V [114]. This can be attributed to the high crystal quality of the CVD-synthesized $MoS_2$ and the $MoS_2$/Si interface. Furthermore, the device demonstrated an unprecedented response speed with a rise and a decay times of 56 and 825 ns, respectively.

Planar Si is not conducive of efficient light absorption; therefore, Si with a nanostructured surface is used to fabricate TMD/Si heterojunction photodetectors. Dhyani et al. prepared nanostructured porous Si through the electrochemical etching method and deposited $MoO_3$ via magnetron sputtering, followed by sulfurization, to fabricate a $MoS_2$/Si heterojunction photodetector [115]. The device demonstrated a maximum responsivity of 9 A/W, a high specific detectivity of $10^{14}$ Jones and a fast response time of 9 µs. It exhibited improved photodetection performance compared with a $MoS_2$/planar Si device, which can be attributed to the higher interfacial barrier height, superior light trapping and larger junction area in the $MoS_2$/porous Si junction. Typically, TMDs synthesized using the thermolysis method exhibit poor crystal quality. To address this issue, Xiao et al. synthesized a reduced graphene oxide (RGO)-$MoS_2$ composite film by the thermolysis of a $(NH_4)_2MoS_4$-graphene oxide (GO) mixture film, followed by annealing in a reducing atmosphere [58]. The RGO-$MoS_2$ film was directly synthesized on a pyramid-Si substrate, and a trilayer graphene film was transferred onto the heterojunction as the transparent top electrode, as shown in Figure 5d. The device exhibited excellent performance, including a large responsivity of 21.8 A/W, an extremely high specific detectivity of $3.8 \times 10^{15}$ Jones and an ultrabroad spectral response ranging from 0.35 to 4.3 µm. The broadband spectral response was attributed to the narrowed band gap of $MoS_2$ resulting from imperfect crystallinity. Wu et al. also reported ultrabroad band photodiodes based on defective $WS_2$ film/pyramid Si heterojunctions [116].

In order to achieve broadband photodetection, Wu et al. employed a type II Weyl semimetal 1T'-$MoTe_2$ as the 2D material to form a vdW heterojunction with Si as the photodetector [49]. The device exhibited an ultrabroad band detection spectrum ranging from UV light to mid-IR light (10.6 µm), as shown in Figure 5e,f. The long-wavelength photoresponse relied on the photogenerated carriers in the 1T'-$MoTe_2$, which could transfer to the Si side through thermionic emission or tunneling through an ultrathin insulator layer.

In a recent study, Huang et al. made an important discovery regarding heterojunctions with the stacking sequence of $WS_2$/$WSe_2$/Si, which can form a unipolar barrier heterojunction. In this configuration, $WS_2$ served as the photon absorber, $WSe_2$ acted as the unipolar barrier, and the p-Si functioned as the collector of photogenerated carrier [117]. The inclusion of the $WSe_2$ layer in the middle not only mitigated the detrimental effects of the substrate, but also established a high-conduction band barrier that effectively filtered out various dark current components while allowing the photocurrent to flow unimpeded. As a result, the device exhibited an impressive photo switching ON/OFF ratio of more than $10^5$, along with a high specific detectivity of $2.39 \times 10^{12}$ Jones and a fast response time of 8.47 ms.

By integrating an ultrathin Si (u-Si) substrate with a 2D material, it becomes possible to fabricate a flexible 2D material/Si heterojunction photodiode. Choi et al. successfully demonstrated an ultra-flexible photodiode based on a 2D-$MoS_2$/Si heterojunction, which exhibited outstanding photodetection performance and remarkable flexibility [118]. In their study, a 15 µm thick u-Si substrate was used. A layer of $MoO_3$ (3 nm) was deposited on the u-Si, followed by sulfurization to convert it into $MoS_2$ using an air pressure plasma-enhanced CVD technique. The resulting flexible photodiode displayed a commendable photoresponse, with a responsivity of 10.07 mA/W and a specific detectivity of $4.53 \times 10^{10}$ Jones. Furthermore, the flexible photodetector exhibited excellent stability

when exposed to external pulsed light, regardless of its bending state. In addition, even after 1000 consecutive bending cycles, the ON and OFF currents showed no significant changes. These results underscore the immense potential of flexible photodiodes in the field of wearable photodetectors.

For better comparison and understanding of the TMD/Si heterojunction photodetectors, the performance parameters of some typical devices are listed in Table 2.

**Table 2.** Summary of properties of TMD/silicon heterojunction devices.

| Materials | Measurement Conditions | R/A·W$^{-1}$ | D*/Jones | EQE | Time (Rise/Down) | Ref. |
|---|---|---|---|---|---|---|
| MoS$_2$/Si | 808 nm/$-2$ V | 0.07 | | | | [62] |
| WS$_2$/Si | 660 nm/$-5$ V | 5.7 | | | 670 µs/998 µs | [104] |
| MoS$_2$/Si | 650 nm/$-2$ V | 11.9 | $2.1 \times 10^{10}$ | | 30.5 µs/71.6 µs | [106] |
| MoTe$_2$/Si | 980 nm/0 V | 0.19 | $6.8 \times 10^{13}$ | 24% | 150 ns/350 ns | [107] |
| MoTe$_2$/Si | 700 nm/0 V | 0.26 | $2 \times 10^{13}$ | | 5 ns/8 ns | [108] |
| MoSe$_2$/Si | 980 nm/0 V | 0.7205 | $10^{13}$ | 91% | 13 µs/35 µs | [109] |
| MoS$_2$/Si | 808 nm/0 V | 0.3 | $10^{13}$ | | 3 µs/40 µs | [111] |
| WS$_2$/Si | 365 nm/0 V | 0.004 | $1.5 \times 10^{10}$ | | 1.1 µs | [112] |
| MoS$_2$/Si | 808 nm/$-2$ V | 0.908 | $1.889 \times 10^{13}$ | | 56 ns/825 ns | [114] |
| MoS$_2$/Si | 550 nm/5 V | 9 | $10^{14}$ | | 9 µs/7 µs | [115] |
| RGO-MoS$_2$/Pyramid Si | 808 nm/0 V | 21.8 | $3.8 \times 10^{15}$ | | 2.8 µs/46.6 µs | [58] |
| WS$_2$/Pyramid Si | 980 nm/0 V | 0.29 | $2.6 \times 10^{14}$ | | 5.2 µs/22.3 µs | [116] |
| 1T'-MoTe$_2$/Si | 980 nm/0 V | 0.526 | $2.17 \times 10^{12}$ | | 1.9 µs/41.5 µs | [49] |
| WS$_2$/WSe$_2$/Si | 405 nm/1.5 V | 3.72 | $2.39 \times 10^{12}$ | 1140% | 8.47ms/7.98ms | [117] |
| MoS$_2$/Si | 850 nm/6 V | 0.01007 | $4.53 \times 10^{10}$ | | 78 µs/76 µs | [118] |

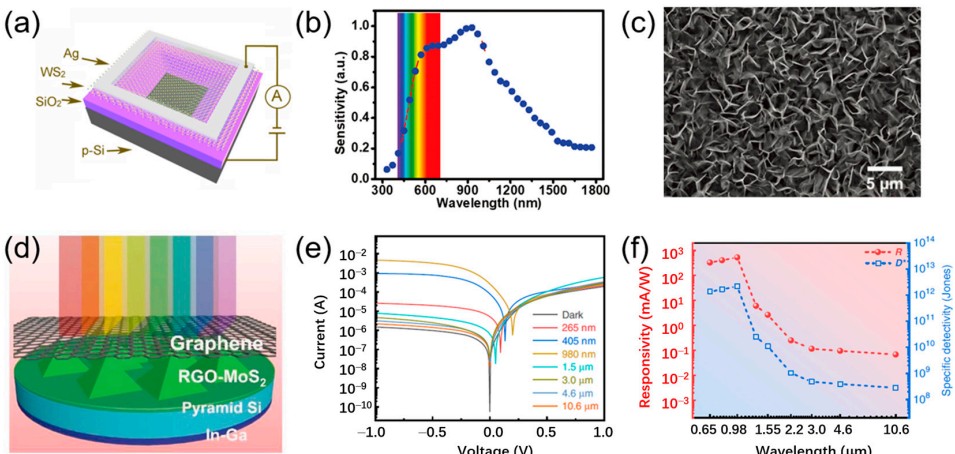

**Figure 5.** Schematic and photodetection performance of TMD/Si heterojunction devices. (**a**) Schematic of WS$_2$/Si heterojunction. Reproduced with permission from ref. [104]. Copyright 2016, American Chemical Society. (**b**) Spectral response of MoTe$_2$/Si heterojunction photodetector. Reproduced with permission from ref. [107]. Copyright 2020, WILEY-VCH. (**c**) Scanning electron microscopy image of vertical aligned MoS$_2$ nanosheets. Reproduced with permission from ref. [113]. Copyright 2017, WILEY-VCH. (**d**) Schematic of the RGO-MoS$_2$/pyramid Si heterojunction photodetector. Reproduced with permission from ref. [58]. Copyright 2018, WILEY-VCH. (**e**) Current–voltage curves with and without light illumination for a 1T'-MoTe$_2$/Si heterojunction photodetector. (**f**) Spectral response and specific detectivity of a 1T'-MoTe$_2$/Si heterojunction photodetector. Reproduced with permission from ref. [49]. Copyright 2023, Nature Publishing group.

### 3.3. NMD-Si Heterojunctions

Recently, NMDs have attracted considerable attention due to their exceptional electrical and optoelectronic properties. For instance, a phototransistor based on few-layer PtS$_2$ demonstrated a remarkably high photoresponsivity of $1.56 \times 10^3$ A/W [119]. PtSe$_2$, on the other hand, displayed a thickness-dependent metal-to-semiconductor transformation [120].



When bilayer $PtSe_2$ was combined with defect modulation, it exhibited strong light absorption in the mid-IR region and showed excellent photodetection performance [121]. Experimental results indicated that the few-layer $PdSe_2$ achieved an impressive field effect mobility of 210 $cm^2$/Vs [122]. The exceptional properties exhibited by these materials make them highly attractive for various photodetection applications.

Aftab et al. employed mechanically exfoliated multilayer n-type $PdSe_2$ nanoflakes to fabricate $PdSe_2$/Si p-n photodiodes [123]. This was achieved by transferring the $PdSe_2$ nanoflakes onto the surface of a p-type Si substrate. To modulate the conductance of the $PdSe_2$ nanoflake in the diode, an ionic liquid gate was utilized. The rectification behavior of the photodiode could be controlled by adjusting the gate voltage, resulting in a maximum rectification ratio of $1.0 \times 10^5$. When exposed to light illumination, the photodiode exhibited exceptional detection performance, including a substantial open-circuit voltage of 0.6 V, a rapid response speed of 17.3 μs and a broadband spectral response ranging from 400 to 1200 nm. In another study, the research same research group investigated the photodetection capability of an n-type $PtS_2$ nanoflake/p-type Si photodiode [124]. To enhance light absorption, the Si substrate was modified with a layer of Si pyramid microstructures Figure 6a. This device demonstrated a robust photovoltaic response to incident light, yielding a significant open-circuit voltage of 0.45 V at a light intensity of 70.32 $mW/cm^2$ at 500 nm. Consequently, the device functioned as a self-powdered detector, boasting a high photoresponsivity of 11.88 A/W.

Exfoliated $PtSe_2$ nanoflakes pose limitations for mass production of devices, thereby limiting their practical applications. To address this, Yim et al. developed a simply synthesis method for $PtSe_2$ by the thermal conversion of pre-deposited Pt films on $SiO_2$/Si substrates [125]. The resulting $PtSe_2$ film was transferred onto an n-type Si substrate, forming a $PtSe_2$/Si Schottky photodiode, as shown in Figure 6b. This photodiode exhibited a pronounced response above the Si band gap, with a maximum responsivity of 490 mA/W, as well as a weaker response blow the band gap ranging from 0.1 to 1.5 mA/W. The photoresponse below the Si band gap was attributed to light absorption in the $PtSe_2$ layer, which exhibited a clear dependence on $PtSe_2$ thickness. In order to enhance light absorption, Zeng et al. introduced a Si nanowire array (SiNWA) to replace the planar Si, thereby creating a $PtSe_2$/SiNWA heterojunction photodiode, as illustrated in Figure 6c [126]. The strong light confinement within the SiNWA resulted in excellent photodetection performance, including a high responsivity of 12.65 A/W, a large specific detectivity of $2.5 \times 10^{13}$ Jones and a fast response time of 19.5 μs. Moreover, due to the broadband light absorption of $PdSe_2$, the photodiode exhibited an ultrabroad spectral response ranging from 0.2 to 1.55 μm. The same research group also observed improved photodetection performance in a $PdSe_2$/SiNWA heterojunction photodiode [127]. Additionally, they discovered that the polycrystalline $PdSe_2$ film/SiNWA heterojunction photodiode displayed polarization-sensitive behavior, with a polarization sensitivity of 75.

To improve the quality of the interface, it is highly preferable to grow a 2D TMD directly on a Si substrate. Xie et al. successfully synthesized a $PtSe_2$ film directly on a Si substrate by the selenization of a Pt film in a CVD chamber at an elevated temperature [128]. Due to the high quality of the $PtSe_2$/Si interface, the resulting photodiode demonstrated excellent performance in both photovoltage and photocurrent modes, with a high voltage and a current responsivity of $5.26 \times 10^6$ V/W and 520 mA/W, respectively, at a wavelength of 808 nm. In another study, Liang et al. directly synthesized a $PdSe_2$ film on a Si substrate with pyramid microstructures, as shown in Figure 6d, to reduce light reflection [129]. The resulting photodiode exhibited outstanding photodetection performance and could function as a self-powered photodetector. When operated at zero bias, the photodiode demonstrated a large photo switch ON/OFF ratio of $1.6 \times 10^5$, a high responsivity of 456 mA/W and a high specific detectivity of $9.97 \times 10^{13}$ Jones at a wavelength of 980 nm. To improve the collection efficiency of photogenerated carriers, multilayer graphene was coated on the surface of a $PtTe_2$/Si heterojunction Figure 6e [130]. Compared to the photodiode without graphene coating, the graphene-coated photodiode displayed improved forward current

and reduced reverse current. Moreover, the addition of the graphene layer also enhanced the photoresponse, resulting in an increased short-circuit current and an open-circuit voltage. In order to enhance the IR response, black phosphorus (BP) QDs were dispersed on the surface of a PdSe$_2$/Si heterojunction, as schematically shown in Figure 6f [131]. The introduction of BP QDs significantly improved the photodetection performance across the entire measured spectral range. The enhanced photoresponse in the middle IR range was attributed to the defects in the BP QDs, which were able to absorb IR light and generate a photocurrent. Ye et al. discovered that the dark current could be greatly reduced by adding an ultrathin layer of SiO$_2$ between PtSe$_2$ and Si [60]. This additional interfacial layer not only reduced both forward and reverse currents but also improved the rectification ratio of the photodiode compared to devices without an SiO$_2$ layer. Additionally, the ideal factor of the photodiode was 1.015, which is close to the ideal value of 1, indicating low interfacial recombination in the PtSe$_2$/ultrathin SiO$_2$/Si heterojunction. Therefore, the photodiode exhibited excellent photodetection performance, including a high responsivity of 8.06 A/W, a high specific detectivity of $4.78 \times 10^{13}$ Jones and an exceptional photo switch ON/OFF ratio of $1.29 \times 10^9$ at zero bias at a wavelength of 808 nm. Furthermore, the responsivity at 1550 nm reached as high as 0.62 mA/W.

For better comparison and understanding of the NMD/Si heterojunction photodetectors, the performance parameters of some typical devices are listed in Table 3.

**Table 3.** Summary of properties of NMD/silicon heterojunction devices.

| Materials | Measurement Conditions | R/A·W$^{-1}$ | D*/Jones | EQE | Time (Rise/Down) | Ref. |
|---|---|---|---|---|---|---|
| PtS$_2$/Si | 500 nm/1 V | 11.88 | | | 2.6 s/2.7 s | [124] |
| PtSe$_2$/Si | 970 nm/$-2$ V | 0.49 | | | | [125] |
| PtSe$_2$/SiNWA | 780 nm/$-5$ V | 12.65 | $2.5 \times 10^{13}$ | | 10.1 μs/19.5 μs | [126] |
| PdSe$_2$/SiNWA | 980 nm/0 V | 0.726 | $3.19 \times 10^{14}$ | | 25.1 μs/34 μs | [127] |
| PtSe$_2$/Si | 880 nm/0 V | 0.52 | $3.26 \times 10^{13}$ | | 55.3 μs/170.5 μs | [128] |
| PdSe$_2$/Pyramid Si | 980 nm/0 V | 0.456 | $9.97 \times 10^{13}$ | 58% | | [129] |
| PdSe$_2$/Si | 780 nm/0 V | 0.3002 | $10^{13}$ | | 38 μs/44 μs (BPQDs@PdSe$_2$/Si) | [131] |
| PtSe$_2$/Si | 808 nm/0 V | 8.06 | $4.78 \times 10^{13}$ | | | [60] |

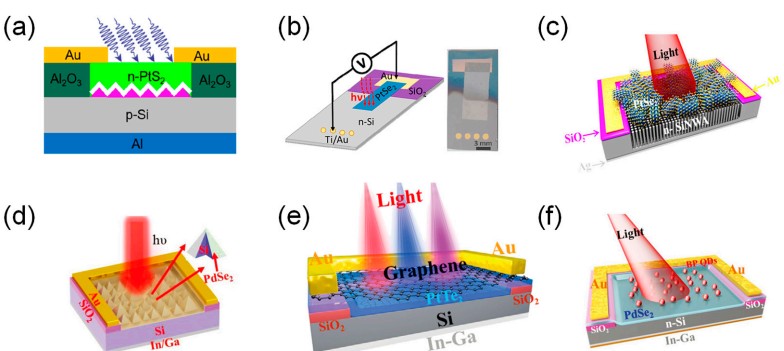

**Figure 6.** NMD/Si heterojunction photodetectors. (**a**) Schematic of the PtS$_2$/pyramid Si heterojunction. Reproduced with permission from ref. [124]. Copyright 2022, American Chemical Society. (**b**) Schematic of the PtSe$_2$/Si heterojunction. Reproduced with permission from ref. [125]. Copyright 2018, American Chemical Society. (**c**) Schematic of the PtSe$_2$/SiNWA heterojunction. Reproduced with permission from ref. [126]. Copyright 2018, Nature Publishing Group. (**d**) Schematic of the PdSe$_2$/pyramid Si heterojunction. Reproduced with permission from ref. [129]. Copyright 2019, WILEY-VCH. (**e**) Schematic of the PtTe$_2$/Si Schottky junction with a graphene top electrode. Reproduced with permission from ref. [130]. Copyright 2020, WILEY-VCH. (**f**) Schematic of the BP QDs-PdSe$_2$/Si heterojunction. Reproduced with permission from ref. [131]. Copyright 2019, WILEY-VCH.

### 3.4. Other 2D/Si Heterojunctions

In addition to graphene, TMDs and NMDs, there are various other types of 2D materials that can form vdW heterojunctions with Si and demonstrate remarkable photodetection capabilities. However, due to the rapid advancements in this field, we will focus on discussing only a few of these materials.

Topological insulators (TIs) are materials characterized by an insulating bulk with spin-momentum-lock Dirac cones on their surfaces. Zhang et al. have predicted that the Tl Dirac-like surface state exhibited strong optical absorption, making them promising candidates for high-performance broadband photodetectors operating from the IR to the terahertz range [132]. Some TIs, such as $Bi_2Te_3$ and $Bi_2Se_3$, are layered materials capable of forming vdW heterojunctions with bulk materials, thereby enabling the development of intriguing optoelectronic devices. Yao et al. employed pulsed laser deposition to deposit a $Bi_2Te_3$ film on an n-Si, forming a $Bi_2Te_3$/Si heterojunction photodetector [133]. This heterojunction demonstrated pronounced rectification behavior, with a rectification ratio exceeding $5 \times 10^3$. Moreover, it exhibited ultrabroad band photodetection capabilities spanning from the UV to the terahertz range. The photoresponse for photon energies greater than the band gap of Si primarily originated from the photogenerated carriers in the Si. Photon energies below the bulk $Bi_2Te_3$ band edge were attributed to the surface state of the $Bi_2Te_3$, while photon energies between the Si band edge and the bulk $Bi_2Te_3$ band edge were attributed to the bulk state of the $Bi_2Te_3$. In another study, Zhang et al. fabricated a $Bi_2Se_3$/Si heterojunction photodiode, as shown in Figure 7a, by depositing a $Bi_2Se_3$ film on the surface of an n-Si substrate by physical vapor deposition [134]. The strong built-in electrical field at the $Bi_2Se_3$/Si interface greatly facilitated the separation and transport of photogenerated carriers. The photodiode consequently exhibited a high responsivity of 24.28 A/W and a high detectivity of $4.39 \times 10^{12}$ Jones at a wavelength of 808 nm under a bias voltage of $-1$ V. More recently, Li et al. prepared a $Bi_2Se_3$/Si heterojunction with vertically oriented $Bi_2Se_3$ nanoplates on a p-Si substrate, as shown in Figure 7b,c [135]. This unique structure enhanced light absorption and provided more efficient paths for photocarrier transport, resulting in improved photodetection performance. Hong et al. utilized a Si substrate with pyramid microstructures to enhance light absorption and directly deposited $Bi_2Se_3$ on the pyramids using physical vapor deposition [136]. This device achieved a wide spectra response ranging from 635 to 2700 nm, a high specific detectivity of $1.24 \times 10^{11}$ Jones and a fast response speed in the microsecond range.

2D transition metal carbides, known as MXenes, are a promising class of 2D materials that exhibited intriguing electronic and optoelectronic properties [137,138]. Kang et al. conducted a study on the optoelectronic properties of $Ti_3C_2T_x$/Si heterojunction photodetectors [139]. They discovered that the $Ti_3C_2T_x$ not only served as a transparent electrode but also facilitated the separation and transportation of photogenerated carriers. Although the photodetectors exhibited fast response speeds and a wide dynamic range, their responsivity and detectivity were relatively low. This can be attributed to the presence of a high density of surface states between the MXene and the Si, leading to carrier recombination and the Fermi-level pinning of Si. To address this issue, Song et al. achieved high specific detectivity and responsivity in photodiodes by incorporating a chemically regrown interfacial $SiO_x$ layer Figure 7d and precisely controlling the thickness of the $Ti_3C_2$ MXene [140]. These photodiodes demonstrated a high specific detectivity of $2.03 \times 10^{13}$ Jones and an impressive responsivity of 402 mA/W at zero bias and 900 nm, as shown in Figure 7e. To enhance the stability of the $Ti_3C_2$ MXene, a layer of Silicone was applied to the device, resulting no noticeable degradation. Additionally, Liu et al. successfully reduced the dark current in photodiodes by introducing a thin layer of h-BN (sub-1 nm thick) as the interfacial layer, exploiting its dangling bond-free nature [141]. As a result, the dark current was reduced by two orders of magnitude and the specific responsivity reached $10^{13}$ Jones.

Monoelemental 2D materials, such as BP, antimony, bismuth and tellurium, have gained considerable attention in recent decades due to their unique electrical and opto-

electronic properties [142,143]. Tellurium, in particular, has emerged as a focal point of research due to its intriguing characteristics, including a thickness-dependent band gap ranging from 0.3 to 1.04 eV [144,145], a high hole mobility of about 700 cm$^2$/Vs at room temperature [146] and exceptional air stability [147]. Consequently, the integration of Te with Si can lead to the development of heterojunction photodetectors with fascinating photodetection capabilities. In a study by Zeng et al., Te nanosheets were synthesized by a hydrothermal method and subsequently transferred onto the surface of an n-Si substrate to create p-n photodiodes [148]. Type I heterojunction was formed between p-Te and n-Si with a built-in electrical field at the interface. The photodiodes accordingly exhibited low dark current and displayed a robust photoresponse even under zero bias condition, including an impressive responsivity of 6.49 A/W and a high specific detectivity of $7.79 \times 10^{12}$ Jones at a wavelength of 808 nm. Additionally, these devices demonstrated a polarization-sensitive response to linearly polarized light, exhibiting an anisotropic ratio of 2.1 at 635 nm due to the in-plane low-symmetry atomic structure of the Te nanosheets. Nonetheless, the transfer of nanosheets for the fabrication of Te/Si heterojunctions is unsuitable for the mass production of devices. In addition, the contamination of the interface is difficult to avoid during the transfer process. To address this issue, Lu et al. developed Te/Si heterojunction arrays by directly depositing Te films onto pre-patterned SiO$_2$/Si substrates using a PLD technique [66]. Utilizing the small band gap of the Te, the resulting photodiodes exhibited an ultrabroad spectral response extending from the UV to the near-IR (370.6–2240 nm). Moreover, the substantial built-in electrical field in the interface enabled the photodiodes to achieve a high photo switching ON/OFF ratio of $10^8$ and a rapid response speed. Furthermore, a matrix array of $35 \times 35$ devices was fabricated, showcasing an average responsivity of 249 A/W, an EQE of 76,350% and a specific detectivity of $1.15 \times 10^{11}$ Jones. Similarly, Li et al. fabricated Te/Si heterojunctions by magnetron sputtering a Te film onto the surface of a Si substrate, as shown in Figure 7f [61]. Although the device did not exhibit obvious rectification, it delivered excellent photodetection performance, including an ultrahigh responsivity of 437.24 A/W and a high specific detectivity of $4.86 \times 10^{11}$ Jones at 1064 nm under zero bias. In addition, the device had a fast response speed, with a rise time of 920 ns and a decay time of 200 µs.

For better comparison and understanding of the other 2D material/Si heterojunction photodetectors, the performance parameters of some typical devices are listed in Table 4.

**Table 4.** Summary of properties of Other 2D material/silicon heterojunction devices.

| Materials | Measurement Conditions | R/A·W$^{-1}$ | D*/Jones | EQE | Time (Rise/Down) | Ref. |
|---|---|---|---|---|---|---|
| Bi$_2$Te$_3$/Si | 635 nm/0 V<br>635 nm/−5 V | 0.017<br>1 | $2.5 \times 10^{11}$ | | | [133] |
| Bi$_2$Se$_3$/Si | 808 nm/0 V<br>808 nm/−1 V | 2.6<br>24.28 | $4.39 \times 10^{12}$<br>$1.21 \times 10^{12}$ | | 2.5 µs/5.5 µs | [134] |
| Bi$_2$Se$_3$/Pyramid Si | 1550 nm/0 V<br>2700 nm/0 V | $3.06 \times 10^{-8}$<br>$1.8 \times 10^{-8}$ | $1.37 \times 10^5$<br>$1.53 \times 10^6$ | | 0.52 ms/0.44 ms<br>0.585 ms/0.535 ms | [136] |
| Ti$_3$C$_2$/Si | 910 nm/0 V | 0.402 | $2.3 \times 10^{13}$ | 60.3% | 0.14 ms/1.6 ms | [140] |
| Te/Si | 808 nm/0 V | 6.49 | $7.79 \times 10^{12}$ | 998% | 26 ms/30 ms | [148] |
| Te/Si | 405 nm/−2 V | 249 | $1.15 \times 10^{11}$ | 76,350% | 3.7 ms/4.4 ms | [66] |
| Te/Si | 1064 nm/0 V | 437.24 | $4.86 \times 10^{11}$ | | 920 ns/200 µs | [61] |

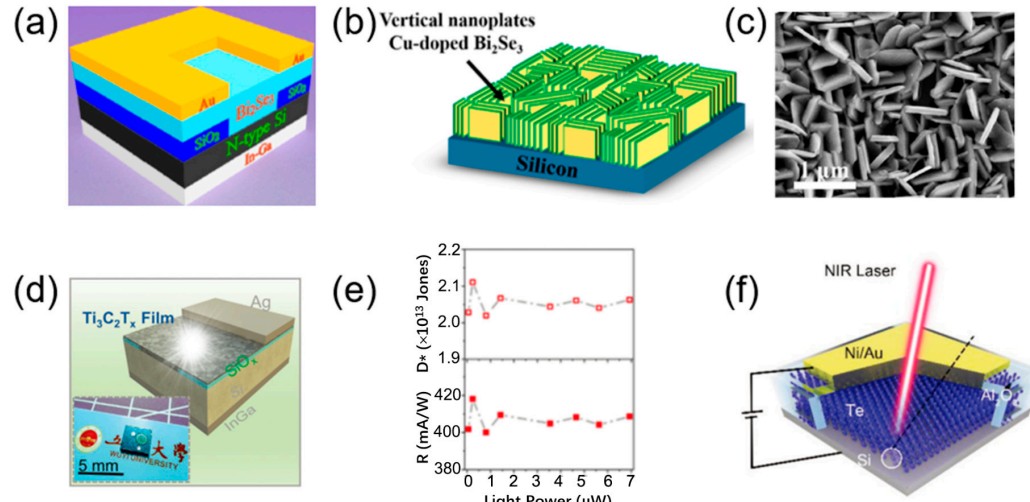

**Figure 7.** Other 2D material/Si heterojunction photodetectors. (**a**) Schematic of the $Bi_2Se_3$/Si heterojunction. Reproduced with permission from ref. [134]. Copyright 2016, American Chemical Society. (**b**) Schematic of the vertically aligned $Bi_2Se_3$ nanoplates/Si heterojunction. (**c**) Scanning electron microscopy image of the vertically aligned $Bi_2Se_3$ nanoplates. (**b**,**c**) Reproduced with permission from ref. [135]. Copyright 2020, American Chemical Society. (**d**) Schematic of the $Ti_3C_2$/$SiO_x$/Si heterojunction and digital photograph of the device. (**e**) Responsivity and specific detectivity as a function of light power for the $Ti_3C_2$/$SiO_x$/Si heterojunction photodiode. (**d**,**e**) Reproduced with permission from ref. [140]. Copyright 2021, WILEY-VCH. (**f**) Schematic of the Te/Si heterojunction. Reproduced with permission from ref. [61]. Copyright 2023, WILEY-VCH.

## 4. Summary and Outlook

In summary, significant progress has been made in enhancing the photodetection performance of 2D material/Si heterojunction photodetectors. The performance of these photodetectors surpasses that of commercial Si photodiodes in certain aspects, including detection spectral range, detectivity and responsivity. However, several challenges still hinder the practical application of 2D material/Si heterojunction photodetectors, which are listed below.

1.  The gain mechanism in semiconducting 2D material/Si heterojunction remains unknown. Further experimental and theoretical investigations are necessary to elucidate the unique gain mechanism in these heterojunctions.
2.  Polarization-sensitive photodetectors offer more information compared to isotropic photodetectors. However, Si, being an isotropic material, lacks polarization that is sensitive to polarized light. Some 2D materials possess intrinsic anisotropy, making it possible to achieve a polarization-sensitive 2D material/Si heterojunction photodetector by employing anisotropic 2D materials. Nevertheless, the anisotropic property cannot be maintained in polycrystalline 2D material films, and synthesizing large-area single crystal 2D materials remains a significant challenge. Alternatively, utilizing optical structures to achieve polarization-sensitive photodetection is a viable approach.
3.  Most graphene/Si heterojunctions rely on transferring graphene onto Si, which unavoidably introduces residues and defects at the interface, leading to performance degradation. Exploring alternative methods, such as the direct growth of graphene on Si, can alleviate this issue.
4.  The crystal quality of directly grown 2D materials on Si substrates is poor. These 2D materials are polycrystalline with numerous defects. These defects can act as recombination centers, compromising the performance of the photodetectors. Therefore, it is crucial to exploring new techniques to enhance the quality of 2D materials. Additionally, for integration with COMS readout circuits, synthesis methods compatible with COMS technology are required.

5. The controlled doping of 2D materials is necessary to regulate the built-in electrical field and depletion region, optimizing the performance of the heterojunction photodetectors. However, achieving stable and reliable doping in 2D materials remains a challenge due to their ultrathin thickness.

6. Most reported 2D material/Si heterojunction photodetectors exhibit exceptional key parameters in one or two aspects, which may not be suitable for practical applications. Photodetectors with well-balanced key parameters are preferable. Therefore, it is essential to comprehensively evaluate the key parameters of photodetectors.

Si has well-established processing technology within the optoelectronic field, while heterojunctions fully exploits the advantages of both the 2D materials and the Si. In addition to the above six points, numerous challenges still impede the commercialization of 2D material/Si heterojunction photodetectors. However, through ongoing research on these photodetectors, solutions can be found for growth, device fabrication, and mass production concerns. This progress will be accompanied by the exploration of novel detection mechanisms, innovative ideas and advancements in device structures.

**Author Contributions:** Conceptualization, C.L. (Changyong Lan); investigation, Y.W. and S.Z.; visualization, Y.W.; writing—original draft preparation, Y.W.; writing—review and editing, Y.W., C.L. (Changyong Lan) and C.L. (Chun Li); supervision, C.L. (Changyong Lan); project administration, C.L. (Changyong Lan); funding acquisition, C.L. (Changyong Lan) and C.L. (Chun Li). All authors have read and agreed to the published version of the manuscript.

**Funding:** This research was funded by the National Natural Science Foundation of China (Grant Nos. 62074024 and 61975024), the Sichuan Science and Technology Program (Grant Nos. 2023YFH0090 and 2023NSFSC0365), the Natural Science Foundation of Sichuan Province (Grant No. 2022NSFSC0042) and the National Key Research and Development Program of China (Grant No. 2019YFB2203504).

**Institutional Review Board Statement:** Not applicable.

**Informed Consent Statement:** Not applicable.

**Data Availability Statement:** Not applicable.

**Conflicts of Interest:** The authors declare no conflict of interest.

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
