# Peer review of "Recent Advances in Photodetectors Based on Two-Dimensional Material/Si Heterojunctions"

_applsci, doi:10.3390/app131911037_

Round 1

Reviewer 1 Report

Dear Editor,

Review

 I examined "the Recent Advances in Photodetectors Based on Two-Dimensional 2 Material/Si Heterojunctionstitled "Review. It includes research on the electronic and optoelectronic properties of two-dimensional (2D) materials. It reviews recent advances in the field of 2D material/Si heterojunction photodetectors, including advances in photodetector-based 2D material/Si heterojunctions and key performance measurements to evaluate photodetector performance.

The review acceptable with minor corrections.

The  English needs a slight revision. Long sentences should be shorter and more understandable.

1.      It contains information taken from some studies on Two-Dimensional 2 Material/Si Heterojunctions, generally between 2004 and 2022. The number of studies conducted in 2023 is little.

It will be further enriched if the following articles from 2023 are added to the introduction section of the review.

 DOI: 10.3390/coatings13061088,

DOI 10.1088/1361-6463/acb6a3

DOI: 10.3390/photonics10070780

 DOI: 10.26599/NRE.2023.9120058

DOI: 10.1039/D3RA03104G

DOI: 10.3390/nano13081379

DOI: 10.1016/j.mtphys.2023.101042

DOI: 10.29026/oea.2023.230077

DOI: 10.1002/smll.202205778

DOI: 10.1007/s12274-023-5942-1

 DOI: 10.1021/acsaelm.1c01349

 DOI: 10.1038/s41378-023-00548-6

DOI: 10.3389/fphy.2023.1150684

DOI: 10.1007/s40820-023-01048-y

DOI: 10.1063/5.0163938

DOI: 10.34133/adi.0006

2.     Abstract can be improved.

3.     The   axis writings in Figures 2-7 should be clearer.

4.     There are deficiencies in the literature information below:

[4] E.V. Castro, K.S. Novoselov, S.V. Morozov, N.M.R. Peres, J. Dos Santos, J. Nilsson, F. Guinea, A.K. Geim, A.H. Castro Neto. Biased bilayer graphene: Semiconductor with a gap tunable by the electric field effect. Physical Review Letters 2007,99

[11] K.F. Mak, C. Lee, J. Hone, J. Shan, T.F. Heinz. Atomically Thin MoS2: A New Direct-Gap Semiconductor. Physical Review Letters 2010,105

[19] J.D. Zhou, J.H. Lin, X.W. Huang, Y. Zhou, Y. Chen, J. Xia, H. Wang, Y. Xie, H.M. Yu, J.C. Lei, D. Wu, F.C. Liu, Q.D. Fu, Q.S. Zeng,  C.H. Hsu, C.L. Yang, L. Lu, T. Yu, Z.X. Shen, H. Lin, B.I. Yakobson, Q. Liu, K. Suenaga, G.T. Liu, Z. Liu. A library of atomically  thin metal chalcogenides. Nature 2018,556,355-+.

[20] X.L. Kang, C.Y. Lan, F.Z. Li, W. Wang, S. Yip, Y. Meng, F. Wang, Z.X. Lai, C.T. Liu, J.C. Ho. Van der Waals PdSe2/WS2 870 Heterostructures for Robust High-Performance Broadband Photodetection from Visible to Infrared Optical Communication  Band. Advanced Optical Materials 2021,9.

[22] C.Y. Lan, D.P. Li, Z.Y. Zhou, S.P. Yip, H. Zhang, L. Shu, R.J. Wei, R.T. Dong, J.C. Ho. Direct Visualization of Grain Boundaries  in 2D Monolayer WS2 via Induced Growth of CdS Nanoparticle Chains. Small Methods 2019,3.

[25] X.Y. Jia, C.Y. Lan, C. Li. Recent advances in two-dimensional materials in infrared photodetectors (invited). Infrared and Laser  Engineering 2022,51,16. 882

[26] S.J. Liang, B. Cheng, X.Y. Cui, F. Miao. Van der Waals Heterostructures for High-Performance Device Applications: Challenges and Opportunities. Advanced Materials 2020,32.

[27] Q.Y. Tang, F. Zhong, Q. Li, J.L. Weng, J.Z. Li, H.Y. Lu, H.T. Wu, S.N. Liu, J.C. Wang, K. Deng, Y.L. Xiao, Z. Wang, T. He. Infrared Photodetection from 2D/3D van der Waals Heterostructures. Nanomaterials 2023,13. Yayın bilgileri eksik

[30] L. Peng, L.X. Liu, S.C. Du, S.C. Bodepudi, L.F. Li, W. Liu, R.C. Lai, X.X. Cao, W.Z. Fang, Y.J. Liu, X.Y. Liu, J. Lv, M. Abid, J.X. Liu, 891 S.Y. Jin, K.F. Wu, M.L. Lin, X. Cong, P.H. Tan, H.M. Zhu, Q.H. Xiong, X.M. Wang, W.D. Hu, X.F. Duan, B. Yu, Z. Xu, Y. Xu, C.  Gao. Macroscopic assembled graphene nanofilms based room temperature ultrafast mid-infrared photodetectors. Infomat 2022,4.

[31] D. Wu, C.G. Guo, L.H. Zeng, X.Y. Ren, Z.F. Shi, L. Wen, Q. Chen, M. Zhang, X.J. Li, C.X. Shan, J.S. Jie. Phase-controlled van der 894 Waals growth of wafer-scale 2D MoTe2 layers for integrated high-sensitivity broadband infrared photodetection. Light-Science 895 & Applications 2023,12

[32] R.H. Bube.Photoelectronic Properties of Semiconductors;Cambridge University Press:Cambridge,1992. 897

[33] S.O. Kasap.Optoelectronics and Photonics: Principles and Practices,2nd ed;Pearson Education:New Jersey,2013.

[35] M. Garin, J. Heinonen, L. Werner, T.P. Pasanen, V. Vahanissi, A. Haarahiltunen, M.A. Juntunen, H. Savin. Black-Silicon Ultraviolet Photodiodes Achieve External Quantum Efficiency above 130%. Physical Review Letters 2020,125.

[36] P. Xiao, J. Mao, K. Ding, W.J. Luo, W.D. Hu, X.J. Zhang, X.H. Zhang, J.S. Jie. Solution-Processed 3D RGO-MoS2/Pyramid Si Heterojunction for Ultrahigh Detectivity and Ultra-Broadband Photodetection. Advanced Materials 2018,30.

[39] L.L. Li, H. Xu, Z.X. Li, L.C. Liu, Z. Lou, L.L. Wang. CMOS-Compatible Tellurium/Silicon Ultra-Fast Near-Infrared Photodetector. Small 2023.

[40] Y. Li, C.Y. Xu, J.Y. Wang, L. Zhen. Photodiode-Like Behavior and Excellent Photoresponse of Vertical Si/Monolayer MoS2  Heterostructures. Scientific Reports 2014,4.

[57] X. Liu, X.W. Zhang, Z.G. Yin, J.H. Meng, H.L. Gao, L.Q. Zhang, Y.J. Zhao, H.L. Wang. Enhanced efficiency of graphene-silicon  Schottky junction solar cells by doping with Au nanoparticles. Applied Physics Letters 2014,105.

[62] J.J. Zhao, H. Liu, L.E. Deng, M.Y. Bai, F. Xie, S. Wen, W.G. Liu. High Quantum Efficiency and Broadband Photodetector Based  on Graphene/Silicon Nanometer Truncated Cone Arrays. Sensors 2021,21.

[67] A. Di Bartolomeo, F. Giubileo, G. Luongo, L. Iemmo, N. Martucciello, G. Niu, M. Fraschke, O. Skibitzki, T. Schroeder, G. Lupina. Tunable Schottky barrier and high responsivity in graphene/Sinanotip optoelectronic device. 2d Materials 2017,4

R.B. Xiao, C.Y. Lan, Y.J. Li, C. Zeng, T.Y. He, S. Wang, C. Li, Y. Yin, Y. Liu. High Performance Van der Waals Graphene-WS2-Si  Heterostructure Photodetector. Advanced Materials Interfaces 2019,6.

K.E. Chang, T.J. Yoo, C. Kim, Y.J. Kim, S.K. Lee, S.Y. Kim, S. Heo, M.G. Kwon, B.H. Lee. Gate-Controlled Graphene-Silicon  Schottky Junction Photodetector. Small 2018,14.

S. Manzeli, D. Ovchinnikov, D. Pasquier, O.V. Yazyev, A. Kis. 2D transition metal dichalcogenides. Nature Reviews Materials 2017,2.

[75] C. Yim, M. O'Brien, N. McEvoy, S. Riazimehr, H. Schafer-Eberwein, A. Bablich, R. Pawar, G. Iannaccone, C. Downing, G. Fiori, M.C. Lemme, G.S. Duesberg. Heterojunction Hybrid Devices from Vapor Phase Grown MoS2. Scientific Reports 2014,4

[78] S. Aftab, M.F. Khan, K.A. Min, G. Nazir, A.M. Afzal, G. Dastgeer, I. Akhtar, Y. Seo, S. Hong, J. Eom. Van der Waals heterojunction  diode composed of WS2 flake placed on p-type Si substrate. Nanotechnology 2018,29

[80] Z.J. Lu, Y. Xu, Y.Q. Yu, K.W. Xu, J. Mao, G.B. Xu, Y.M. Ma, D. Wu, J.S. Jie. Ultrahigh Speed and Broadband Few-Layer MoTe2/Si  2D-3D Heterojunction-Based Photodiodes Fabricated by Pulsed Laser Deposition. Advanced Functional Materials 2020,30.

[81] W.Y. Lei, G.W. Cao, X.K. Wen, L. Yang, P.Z. Zhang, F.W. Zhuge, H.X. Chang, W.F. Zhang. High performance MoTe 2/Si  heterojunction photodiodes. Applied Physics Letters 2021,119.

[83] H.Y. Jang, J.H. Nam, J. Yoon, Y. Kim, W. Park, B.J. Cho. One-step H2S reactive sputtering for 2D MoS2/Si heterojunction photodetector. Nanotechnology 2020,31.

[86] R.D. Cong, S. Qiao, J.H. Liu, J.S. Mi, W. Yu, B.L. Liang, G.S. Fu, C.F. Pan, S.F. Wang. Ultrahigh, Ultrafast, and Self-Powered Visible-Near-Infrared Optical Position-Sensitive Detector Based on a CVD-Prepared Vertically Standing Few-Layer MoS2/Si Heterojunction. Advanced Science 2018,5.

[88] V. Dhyani, P. Dwivedi, S. Dhanekar, S. Das. High performance broadband photodetector based on MoS2/porous silicon  heterojunction. Applied Physics Letters 2017,111.

[92] L. Li, W.K. Wang, Y. Chai, H.Q. Li, M.L. Tian, T.Y. Zhai. Few-Layered PtS2 Phototransistor on h-BN with High Gain. Advanced  Functional Materials 2017,27.

[93] A. Ciarrocchi, A. Avsar, D. Ovchinnikov, A. Kis. Thickness-modulated metal-to-semiconductor transformation in a transition metal dichalcogenide. Nature Communications 2018,9.

[94] X.C. Yu, P. Yu, D. Wu, B. Singh, Q.S. Zeng, H. Lin, W. Zhou, J.H. Lin, K. Suenaga, Z. Liu, Q.J. Wang. Atomically thin noble metal  dichalcogenide: a broadband mid-infrared semiconductor. Nature Communications 2018,9.

[95] Y.D. Zhao, J.S. Qiao, Z.H. Yu, P. Yu, K. Xu, S.P. Lau, W. Zhou, Z. Liu, X.R. Wang, W. Ji, Y. Chai. High-Electron- Mobility and  Air-Stable 2D Layered PtSe2 FETs. Advanced Materials 2017,29.

[102] F.X. Liang, X.Y. Zhao, J.J. Jiang, J.G. Hu, W.Q. Xie, J. Lv, Z.X. Zhang, D. Wu, L.B. Luo. Light Confinement Effect Induced Highly  Sensitive, Self-Driven Near-Infrared Photodetector and Image Sensor Based on Multilayer PdSe2/Pyramid Si Heterojunction. Small 2019,15. 1043

 [103] L.H. Zeng, D. Wu, J.S. Jie, X.Y. Ren, X. Hu, S.P. Lau, Y. Chai, Y.H. Tsang. Van der Waals Epitaxial Growth of Mosaic-Like 2D Platinum Ditelluride Layers for Room-Temperature Mid-Infrared Photodetection up to 10.6 um. Advanced Materials 2020,32. 1045

 [104] L.H. Zeng, D. Wu, S.H. Lin, C. Xie, H.Y. Yuan, W. Lu, S.P. Lau, Y. Chai, L.B. Luo, Z.J. Li, Y.H. Tsang. Controlled Synthesis of 1046 2D Palladium Diselenide for Sensitive Photodetector Applications. Advanced Functional Materials 2019,29. 1047 [105] X.A. Zhang, J. Wang, S.C. Zhang. Topological insulators for high-performance terahertz to infrared applications. Physical Review  B 2010,82.

[109] X. Hong, J. Shen, X.Y. Tang, Y. Xie, M. Su, G.J. Tai, J. Yao, Y.C. Fu, J.Y. Ji, X.Q. Liu, J. Yang, D.P. Wei. High-performance broadband photodetector with in-situ-grown Bi2Se3 film on micropyramidal Si substrate. Optical Materials 2021,117. 1057

[110] S. Ali, A. Raza, A.M. Afzal, M.W. Iqbal, M. Hussain, M. Imran, M.A. Assiri. Recent Advances in 2D-MXene Based Nanocomposites for Optoelectronics. Advanced Materials Interfaces 2022,9.

[111] Z.X. Liu, H.N. Alshareef. MXenes for Optoelectronic Devices. Advanced Electronic Materials 2021,7.

 [112] Z. Kang, Y.A. Ma, X.Y. Tan, M. Zhu, Z. Zheng, N.S. Liu, L.Y. Li, Z.G. Zou, X.L. Jiang, T.Y. Zhai, Y.H. Gao. MXene-Silicon Van Der Waals Heterostructures for High-Speed Self-Driven Photodetectors. Advanced Electronic Materials 2017,3.

[113] W.D. Song, Q. Liu, J.X. Chen, Z. Chen, X. He, Q.G. Zeng, S.T. Li, L.F. He, Z.T. Chen, X.S. Fang. Interface Engineering Ti3C2 MXene/Silicon Self-Powered Photodetectors with High Responsivity and Detectivity for Weak Light Applications. Small 2021,17.

[115] Z.Y. Lin, C. Wang, Y. Chai. Emerging Group-VI Elemental 2D Materials: Preparations, Properties, and Device Applications.  Small 2020,16.

[118] Z. Shi, R. Cao, K. Khan, A.K. Tareen, X.S. Liu, W.Y. Liang, Y. Zhang, C.Y. Ma, Z.N. Guo, X.L. Luo, H. Zhang. Two-Dimensional Tellurium: Progress, Challenges, and Prospects. Nano-Micro Letters 2020,12.

5.     Writing the DOI information of the literatures will facilitate access.

6.     Some formulas are not given reference numbers. It would be more appropriate to give it.

7.     Line 207, eq(8)  shoul be as eq.(8)

8.     The "consequently word"  is used 11 times. In some sentences, the word therefore can be used instead of consequently.

9.     A better comparison will be possible if the features of photodetectors according to their type are given in a table. A comparison table is necessary to get clear information about what is the best detector material and its features.

10.  Literature should be given to the definition of Quantum Efficiency and Gain.

 The  English needs a slight revision. Long sentences should be shorter and more understandable.

Reviewer 2 Report

overall manuscript is improved..

No issue with that

Author Response

Thank you for your positive comments on our manuscript. We have tried our best to improve the quality of the manuscript according to the reviewers's comments.

Reviewer 3 Report

Two-dimensional (2D) materials and their associated van der Waals heterojunctions have become the subject of extensive research in the fields of materials science, condensed matter physics, electronics, and optoelectronics. Although several review papers have been published on 2D materials and related van der Waals heterojunctions, there is a significant demand for a review paper specifically focusing on 2D material/Si heterojunction photodetectors, given the rapid progress in this area. This paper provides a comprehensive review of the recent progress in 2D/Si heterojunctions, which I believe is both timely and capable of attracting a wide readership. Before it can be accepted for publication, I have a few questions and suggestions:

1. Regarding the gain mechanism in graphene/Si heterojunction photodetectors, it is crucial to include a recent study (Light Sci. Appl., 2021, 10, 113) that demonstrated the impact ionization resulting from a high electrical field in the interfacial layer. This gain mechanism should be incorporated into the review paper.

2. To facilitate better comparison, it is recommended to present the performance data of the photodetectors from literature in tabular form.

3. Please carefully review the paper to identify and rectify any typing mistakes.

4. Some research works related to this study (Nano Lett. 2023, 23, 8241-8248; Adv. Funct. Mater., 2018, 28, 1705970; Adv. Sci. 2019, 6, 1901134; IEEE Trans. Electron Dev. 2022, 11, 6212-6216; ACS Nano, 2019, 13, 9907-9917; ACS Nano, 2021, 15, 10119-10129; ACS Nano, 2022, 16, 5545-5555) should be cited.

No

Reviewer 4 Report

Authors are encouraged to suggested minor changes.

Recent Advances in Photodetectors Based on Two-Dimensional Material/Si Heterojunctions

The review highlights the role of 2D material/Si heterojunctions and their application in heterojunctions. Section 2 provides the basic background information of photodetectors. The figures used from relevant literature are promptly placed. Overall the review is well presented and organized. However, minor corrections are still needed.

1.     Line 25: 2D already abbreviated in Abstract (Line 9). Abbreviate Si and vdW in abstract. Abbreviations should be used at the first mention. UV is already abbreviated in Line 49and remove in Line 562.

2.     Section 1 introduction and section 4 summary and outlook are too generic and do not provide concise and qualitative information.

3.     Please provide limitations and future research prospects under separate heading.

4.     Include few tables, as literature is available.

5.     References, only 40 references out of 121 are from after 2020. Please include more recently published literature.

Few typographical and grammatical errors needs to be rectified.

Reviewer 5 Report

Authors have done well in preparing the manuscript. Minor changes are needed. 

Recent Advances in Photodetectors Based on Two-Dimensional Material/Si Heterojunctions

Overview

This review clearly presents the photodetectors introduction before going in detail in to the topic. The information provided is good. Sections are properly organized and presented with relevant references.

Major Comments

1.     Introduction should be more specific and please include a table or two in the introduction comparing with other metal heterojunctions and Si.

2.     Please include more recently published articles.

3.     Also provide limitations of the Si heterojunctions in photodetectors.

Minor Comments

1.     Abbreviations should only be used if the intended words are repeated more than thrice.

2.     Please check few typographical mistakes.

Remark

The review is well presented and can be considered after minor revision.

Author Response

#Comment 1:Introduction should be more specific and please include a table or two in the introduction comparing with other metal heterojunctions and Si.

Response:Thank you for your valuable comment. The aim of the paper is to present the recent advances in 2D material/Si heterojunction photodetectors. Although certain parameters of the emerging 2D material/Si heterojunction photodetectors exhibit superiority over conventional Si-based photodetectors, they are not current capable of replacing them. To maintain focus on the main topic, we have chosen not to include tables comparing 2D material/Si heterojunction photodetectors with other metal heterojunctions and Si. However, for better understanding the performance of 2D material/Si heterojunction photodetectors, we have incorporated tables listing the typical device performance of these photodetectors in the revised manuscript.

#Comment 2:Please include more recently published articles.

Response:Thank you for your valuable comment. More recent articles have been cited in the revised manuscript.

#Comment 3:Also provide limitations of the Si heterojunctions in photodetectors.

Response:Thank you for your valuable comment. The limitations of the Si hterojunctions in photodetectors have been mentioned on page 2 of the revised manuscript.

“The band gap of Si is about 1.1 eV, which sets the long wavelength limit (around 1100 nm) for Si-based photodetectors, thus restricting their ability to detect infrared (IR) light [42]. Moreover, Si-based photodetectors exhibit poor performance in the ultraviolet (UV) wavelength range due to strong surface recombination [43]. In addition, the fabrication process for Si p-n and p-i-n photodiodes is complex and costly [44].”

#Comment 4: Abbreviations should only be used if the intended words are repeated more than thrice.

Response: Thank you for your valuable comment. We have checked the manuscript carefully to avoid these mistakes.

#Comment 5: Please check few typographical mistakes.

Response:  Thank you for your valuable comment. We have checked the manuscript carefully to avoid these mistakes.

Reviewer 6 Report

Date: 25-9-2023

Journal: Applied sciences

Title: Recent Advances in Photodetectors Based on Two-Dimensional Material/Si Heterojunctions.

Dear Editor,   

In this review, the authors present the background and motivation of the review. Next, 17 we discuss the key performance metrics for evaluating photodetector performance.

Originality: strong

Clarity of Presentation: strong

Importance to Field: strong

Comments for authors:

1.     The abstract: is very short. It should introduce more details. Some corrections are highlighted in pdf file

2.     The introduction :has some very long paragraphs. Some corrections are highlighted in pdf file. adjust all references in all review according to style of journal [   ] beside the paragraph not above words.

3.     The whole body of review: Some corrections are highlighted in pdf file

4.      English editing is moderate recommended.

5.      References: must be adjusted according to style of the journal

6.      Graphical Abstract: I recommend to add graphical abstract summarize the whole idea of paper.

7.     Highlights: authors should add  

1.     The main question addressed by the research

The main idea of this review is clear and focus on significant progress has been made in enhancing the photodetection a performance of 2D material/Si heterojunction photodetectors and authors explained it well.

2. The topic of this review is original and relevant in the field.

3. The Summary and outlook are consistent with the evidence and arguments presented

and they address the main question posed

6. The references are appropriate

7. The tables and figures are appropriate and excellent

Generally, the review needs to revise well especially for typing, grammatical, and

formatting mistakes, requires careful linguistic revision by native English speakers
